# Vespakinin-M, a natural peptide from *Vespa magnifica*, promotes functional recovery in stroke mice

Hairong Zhao [1,2,3], Mei Wang [2], Yuan Gao[2], Xiumei Wu[2,3], Huai Xiao[2,3], Dasong Yang[2,3], Furong He[1], Jiaming Lv[1], De Xie[1], Qiang Wang[1], Weidong Liu[1], Jingang Luo[4], Zizhong Yang[2,3], Chenggui Zhang [2,3✉], Jidong Cheng [1✉] & Yu Zhao [2,3✉]

Acute ischemic stroke triggers complex systemic pathological responses for which the exploration of drug resources remains a challenge. Wasp venom extracted from *Vespa magnifica* (Smith, 1852) is most commonly used to treat rheumatoid arthritis as well as neurological disorders. Vespakinin-M (VK), a natural peptide from wasp venom, has remained largely unexplored for stroke. Herein, we first confirmed the structure, stability, toxicity and distribution of VK as well as its penetration into the blood–brain barrier. VK (150 and 300 μg/kg, *i.p.*) was administered to improve stroke constructed by middle cerebral artery occlusion in mice. Our results indicate that VK promote functional recovery in mice after ischemia stroke, including an improvement of neurological impairment, reduction of infarct volume, maintenance of blood-brain barrier integrity, and an obstruction of the inflammatory response and oxidative stress. In addition, VK treatment led to reduced neuroinflammation and apoptosis associated with the activation of PI3K–AKT and inhibition of IκBα–NF-κB signaling pathways. Simultaneously, we confirmed that VK can combine with bradykinin receptor 2 (B2R) as detected by molecular docking, the B2R antagonist HOE140 could counteract the neuro-protective effects of VK on stroke in mice. Overall, targeting the VK–B2R interaction can be considered as a practical strategy for stroke therapy.

[1] School of Medicine, Xiamen University, Xiamen, Fujian, China. [2] Yunnan Provincial Key Laboratory of Entomological Biopharmaceutical R&D, Dali University, Dali, China. [3] National-Local Joint Engineering Research Center of Entomoceutics, Dali, China. [4] Clinical lab, Xiang'an Hospital of Xiamen University, Xiamen, Fujian, China. ✉email: chenggui_zcg@hotmail.com; jidongcheng36@hotmail.com; dryuzhao@126.com

Acute ischemic stroke (AIS) remains a leading cause of disability and mortality, with an increasing incidence and causing serious harm to human health[1,2]. Within 8–10 min of ischemic attack, neurons in the ischemic core area immediately undergo irreversible necrosis and a salvageable penumbra, dominated by neuronal apoptosis, is formed around the ischemic core area[3,4]. Currently, the main therapeutic strategies[5] for AIS focus on restoring cerebral blood flow (CBF) and saving the ischemic penumbra by the administration of thrombolytic drugs such as intravenous recombinant tissue-type plasminogen activator[5–8]. However, when intravenous thrombolysis has a time window up to 4.5 h, <3% of stroke patients are able to benefit from these interventions and two-thirds of patients have different degrees of disability[9]. Subsequently, reperfusion by thrombolytic therapy after AIS can also accelerate cerebral injury, resulting in brain edema, brain hemorrhage, and neuronal death. This phenomenon is termed as cerebral ischemia/reperfusion (I/R) injury and is implicated in various types of cellular stress[10], including energy failure, oxidative stress, elevation of the intracellular $Ca^{2+}$ levels, release of excitatory neurotransmitters, neuroinflammatory response, and apoptosis. The cerebral tissue around the ischemic core area exacerbates progressively within a few days after stroke. Therefore, a working neuroprotective therapy in stroke is still being explored.

Wasp venom extracted from *Vespa magnifica* (Smith, 1852) is most commonly used to treat rheumatoid arthritis[11] as well as neurological disorders, including epilepsy[12], Parkinson disease[13], and Alzheimer disease[14]. Previously[15], we purified and identified four compounds from wasp venom, namely 5-hydroxytryptamine, Vespakinin-M (VK), Mastoparan-M, and Vespid chemotactic peptide-M. VK, a novel bradykinin (BK) analog containing hydroxyproline found in the venom of *Vespa mandarinia* (Smith, 1852), was first reported in 1979 (Kishimura et al.[16]). BK is an endogenous peptide and its related kinins have been implicated as important pathophysiological mediators of pro-inflammatory, pain-producing, and vasoactive properties[17–20], which exert their effects mediated by two main G protein-coupled BK receptors termed kinin B1 (B1R) and kinin B2 (B2R)[21–24]. However, accumulating evidence has also demonstrated that BK prevents the activation of microglia[25] and neuronal death[26,27] and promotes cell migration and neurogenesis[28–30] as well as angiogenesis and cerebral perfusion in experimental stroke[31].

Although the blockade of B1R or B2R contributes to improving spatial learning and memory deficits[20,22,24], the physiological properties of these receptors are also compromised. Thus, the effects of neuroprotection of BK remain contentious. VK is a BK analog; however, no study has yet reported the effects of VK on stroke. Herein, we show that VK can penetrate the blood–brain barrier (BBB) and acts as a neuroprotective agent to orchestrate functional recovery after stroke in mice. We also demonstrate that the observed reductions in neuroinflammation and apoptosis by VK treatment were associated with the activation of PI3K–AKT and inhibition of IκBα–NF-κB signaling pathways. Additionally, the B2R antagonist HOE140 could eliminate the neuroprotective effects of VK on stroke in mice. Thus, targeting the VK–B2R interaction can be considered a practical strategy for stroke therapy.

## Results

All sham-operated mice survived, but the overall mortality of MCAO was 29.06% (Supplementary Tables 1 and 2). The mortality was different among the experimental groups (MCAO: 29.06% Vs VK-300:18.75%) (Supplementary Table 2). All mice by MCAO-induced were grouped according to the severity of the score by the Longa test.

**Structure, stability in plasma, systemic toxicity, and distribution in the brain of VK**. VK was subjected to Edman sequencing, mass spectrometry (MS/MS), and amino sequence acid determination. The MS/MS ion at m/z 1361.6854 was assigned to the quasi-molecular ion$^{3+}$ of VK (Fig. 1a). The amino acid sequence of VK was determined to be Gly-Arg-Pro-Hyp-Gly-Phe-Ser-Pro-Phe-Arg-Ile-Asp. To evaluate the neurotoxicity of VK, the CCK8 assay was firstly performed in BV2 microglia cells or HT22 neuronal cells. VK inhibited the growth of BV2 cells and HT22 cells at concentrations of 73.5–735 μM (Supplementary Fig. 1a–d). Therefore, VK at concentrations of 0.0735–7.35 μM was used for all cell experiments. We next sought to understand whether VK (0.0091–0.294 μM) inhibits platelet aggregation (Supplementary Fig. 2a). Platelet-activating factor (PAF), arachidonic acid (AA), adenosine diphosphate (ADP), thrombin, and collagen (COL) or saline as inducing agents was employed. ADP-induced platelet aggregation was decreased in VK groups as compared to the control group, detailed values are shown in Supplementary Fig. 2a.

To ensure that the safety of VK in vivo, acute toxicity test was performed. Mice were treated intraperitonelly with a single dose of various concentrations (1.5, 6, 24, 96, and 384 mg/kg) of VK and observed for 24 h and no mortality was observed. Clinical symptoms (temperature, change in skin, eye color change, general physique, body weight, diarrhea, sedation, and organ index) were recorded (Supplementary Table 3). Mice were treated with VK (24, 96, and 384 mg/kg) continued to lose weight until day 5 (Supplementary Fig. 2b). The index of brain, myocardium, and kidney had no difference in all treatment dose groups, whereas increased liver index was found (VK, 384 mg/kg) (Supplementary Fig. 2c–f). No obvious change was observed for four coagulation during the study (Supplementary Fig. 2g–j). In the histopathological study, it was observed that in all treated groups after 14 days (Supplementary Fig. 3a–d) the organs (brain, liver, and kidney) showed no changes at the cellular level in comparison to the control. Karyopyknosisa and the cellular swelling of the hepatocyte were moderate in the liver of mice treated with VK (96 and 384 mg/kg). Taken together, this finding reveals these dosages of VK are safe to mice.

Proteolytic degradation of peptide-based drugs is generally considered as the main defect to limit systemic therapeutic applications. Therefore, high-performance liquid chromatography (HPLC) was used to assess the stability of VK in PBS or plasma at physiological pH (7.4) and temperature (37 °C). The results indicated that VK in PBS or plasma was gradually degraded, with the degradation rate of VK in PBS being 62.4% and that in plasma was completely degraded at 120 min (Fig. 1b–d). For neuropharmacology, crossing the BBB is a remarkable challenge, and VK was labeled with FITC in the N-terminus. FITC-labeled VK was then administered to stroke mice. After 30 min, these mice were imaged by an in vivo imaging system. Because the BBB is partially compromised by stroke insults, VK was found to slightly accumulate in the ischemic region (Fig. 1e). Brains were collected at 0, 30, 60, 90, and 120 min and the distribution of VK was observed. FITC-labeled VK was seen in the cortex, hippocampus, and striatum (Fig. 1e), while these areas of the brain suffered from massive neuronal cell death after focal ischemia. To further evaluate whether VK was a cell-penetrating peptide, the oxygen-glucose deprivation/reoxygenation (OGD/R) model simulated I/R in vivo was established in HT22 hippocampal neuron cells and FITC-labeled VK was incubated for 120 min. Live-cell imaging suggested that FITC-labeled VK (7.35 μM) partially crossed the cell membrane for OGD/R-treated (Fig. 1f, Supplementary Fig. 1e, and Supplementary Movie 1), Supplementary Movie 2 as a negative group with FITC. These results indicate that VK can cross the BBB and cell membrane under conditions such as cerebral I/R.

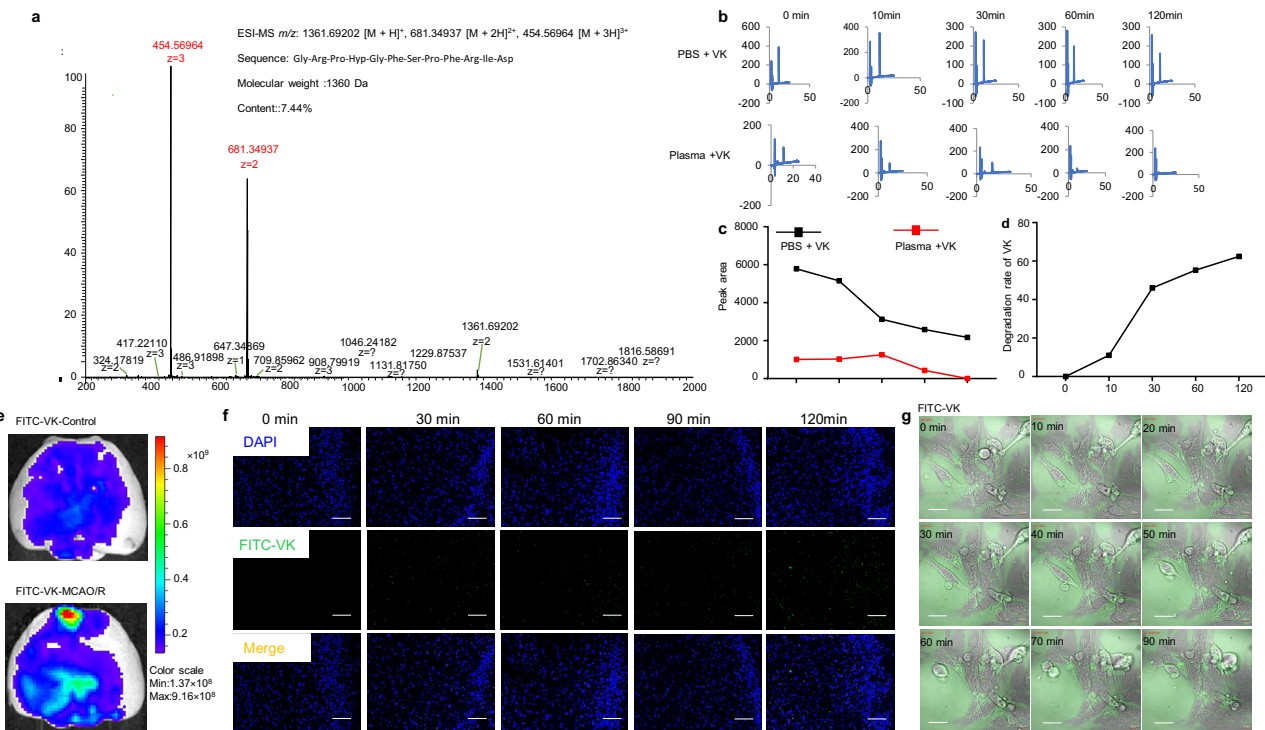

**Fig. 1 Structure, stability, and distribution of VK. a** The matrix-assisted laser desorption/ionization time-of-flight mass spectrometry (MALDI-TOF-MS) profile of Vespakinin-M (VK). The ion at $m/z$ 1361.6854 was the quasi-molecular ion $[M + H]^+$. **b–d** VK (73.5 nM) was incubated for 120 min at physiological pH (7.4) and temperature (37 °C) and the stability of VK in plasma or PBS was continuously detected for 120 min using HPLC. **e** FITC or FITC-labeled VK were administered intravenously stroke mice and they were killed after 30 min to isolate the brain and imaged by an imaging system (IVIS Lumina II), scale bar = 1 mm. As opposed to red, blue represents the lower permeability of VK. **f** Brains were collected and cut after VK treatment at 0, 30, 60, 90, and 120 min. Fluorescent images were acquired by confocal microscopy and the distribution of VK was observed, scale bar = 200 μm. **g** Live-cell imaging was collected in FITC-labeled VK group after OGD/R, scale bar = 20 μm.

## VK alleviates neurological impairment and cerebral infarction during the period of acute cerebral ischemia in mice.

According to the Stroke Therapy Academic Industry Roundtable (STAIR) recommendations[32], we investigated a dose-response experiment of VK treatment on stroke outcomes by middle cerebral artery occlusion/reperfusion (MCAO/R) in mice. In the MCAO/R model, drug treatments and behavior tests were performed according to the methods shown in Fig. 2a. In order to ensure MCAO-induced cerebral ischemia had occurred, laser speckle contrast imaging (LSCI) was used to monitor changes in CBF. CBF was shown to drop to 20–30% of the initial blood flow. During the operation of MCAO, CBF, blood gas, body temperature, and blood pressure were within the normal range, with no difference between the groups. To study the effect of VK on infarct volume, the infarct areas of mice sacrificed at 48 h after MCAO/R were assessed. There was no infarction in the sham group, while the MCAO/R group showed infarction in the striatum, hippocampus, cortex, and caudate nucleus (Fig. 2b, d). VK decreased infarct volume as described in TTC staining (Fig. 2c). Magnetic resonance imaging (MRI) also revealed that VK observably reduced the area of cerebral infarction in MCAO/R mice (Fig. 2d).

The neurological function of mice following MCAO/R was assessed by a series of behavioral tests to observe the effects of VK on sensorimotor and cognitive recovery. The vehicle-treated mice exacerbated deficits in mobility of the left limbs (as shown in the Longa test) as compared to the sham group (Fig. 2e). However, the administration of VK improved neurological deficits compared to vehicle-treated mice (Fig. 2e). In sensorimotor asymmetry assessments, vehicle-treated stroke mice showed a poor grip strength and impaired motor coordination as compared to the sham group (Fig. 2f, g). However, VK treatment improved

functional performance and ameliorated the sensorimotor asymmetry of mice as assessed by the rotarod test and grip test (Fig. 2f, g). Furthermore, to confirm the effects of VK on cognitive function, the classic Morris water maze test was constructed (Fig. 2h–k). Compared with the sham group, an observably increased escape latency to find the platform was found in stroke mice (Fig. 2h); however, VK treatment accelerated the spatial learning ability as observed by a decreased escape latency on day 11–14 (Fig. 2h, j). In the probe trial, VK-treated mice exhibited more crossovers and spent more time in the platform quadrant (spatial memory ability) as compared to the vehicle group (Fig. 2i, k). Collectively, these findings indicate that VK therapy reduces infarct volume and alleviates neurological impairment after MCAO/R in mice.

## VK blocks oxidative stress and restores energy metabolism in the brain after MCAO/R in mice.

Recent data support the view that oxidative stress, including lipid peroxidation and free-radical damage, is a mediator of cerebral I/R injury. In addition, BK (but not VK) shows antioxidant action in cardiovascular disease[33–35]. To evaluate the effects of VK on anti-oxidative damage in stroke mice, we measured markers of lipid peroxidation levels and free-radical damage levels in the brain cortex after a stroke at 48 h (Fig. 3a). Compared with the sham group, the activity of superoxide dismutase (SOD) was reduced in vehicle-treated stroke mice whereas lipid peroxide (LPO) and malondialdehyde (MDA) levels were increased; VK enhanced SOD activity and decreased content of MDA and LPO (Fig. 3b–d). Interestingly, for glutathione, no obvious difference was found between groups (Fig. 3e). To further explain whether VK inhibited oxidative stress

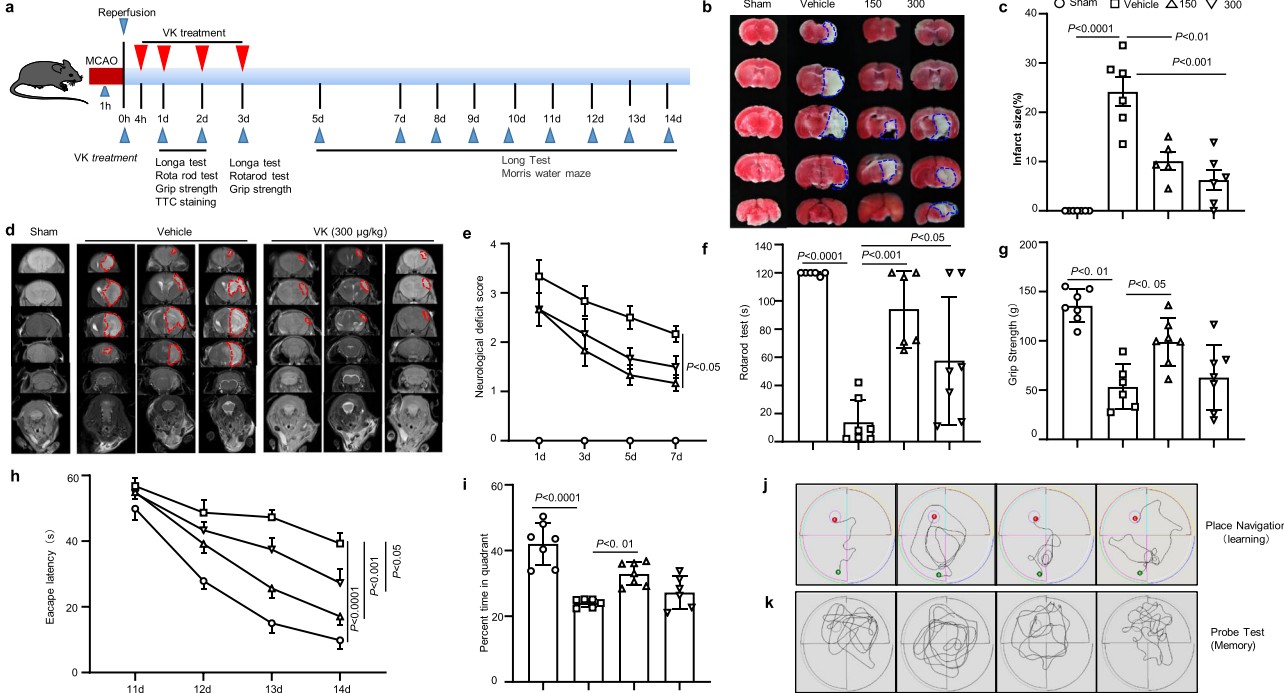

**Fig. 2 VK reduced infract volume and promoted sensorimotor and cognitive function in mice subjected to middle cerebral artery occlusion/reperfusion (MCAO/R).** The administration of the VK schedule and behavioral assessment timeline were illustrated schematically (**a**). Mice (8–10 weeks, males) underwent MCAO for 60 min. The occlusion and reperfusion were confirmed by laser speckle contrast imaging (LSCI). Mice were randomly divided into four groups: sham group, vehicle group, and VK (150 and 300 μg/kg) groups. Mice were administered VK at 0, 4, 22.5, and 46.5 h after MCAO/R as assessed by triphenyl tetrazolium chloride (TTC) staining (**b**, **c**) and MRI (**d**). VK treatment improved sensorimotor recovery as evaluated by the Longa test (**e**), the rotarod test (**f**), and the grip test (**g**) after MCAO/R. The performance in the rotarod test was expressed as the time spent on the rotating rod before falling off and the performance on the grip test was expressed as the score of pullup time for each mouse. Spatial learning and memory were evaluated 11–15 days after MCAO/R by the Morris water maze test. Representative traces indicate the sample paths of the mice from the maze latency trials (learning) (**h**) and the swimming traces from probe trials (memory) (**i**). **j** The latency until the mice located the submerged platform as tested on days 11–14 (defined as spatial learning). **k** Spatial memory was assessed on day 15 by measuring the time spent swimming in the target quadrant. Statistical analyses were performed by Kruskal–Wallis test with the Dunn post hoc test or two-way repeated-measures ANOVA followed by the Bonferroni post hoc test. Data are expressed as means ± SD, $n = 5$–7.

of neurons, we used a reactive oxygen (ROS) kit (DCFH-DA). ROS levels were increased in neurons exposed to the OGD/R model as compared with untreated cells or controls (Fig. 3g, h). Moreover, 24-h pretreatment with VK could rescue the OGD-induced ROS levels, which suggests that VK directly restrained oxidative stress in HT22 cells.

Compounds capable of maintaining energy dynamics in the ischemic penumbra may be attractive therapeutics for stroke therapy. Therefore, kits tested adenosine triphosphate synthase (ATPase) and lactic acid (LD). An increased ATP activity was seen in the VK-treated groups compared with the vehicle-treated group (Fig. 3e). Additionally, the sham group and VK group had a lower LD content compared to the vehicle-treated group (Fig. 3f). These findings suggest that VK could increase energy production in neural cells by promoting mitochondrial ATP production. Given the link between neuronal metabolism and cell function[36], OGD/R model was established to reveal the role of VK on neuron bioenergetics. The dose-dependent stimulation with VK decreased glycolysis (extracellular acidification rate) and accelerated the mitochondrial oxygen consumption rate in neurons (Fig. 3i–l). These results indicate that VK blocks oxidative stress and restores energy metabolism in the brain after MCAO/R in mice.

**VK maintains BBB integrity**. BBB destruction during cerebral I/R is an important pathological change that exacerbates cerebral edema and cerebral infarction[37]. With regards to its structure, a destructive BBB was characterized by a dissolved or fractured basement membrane, exfoliated tight joints (TJs) and endotheliocytes (ECs), vacuolar or swelled mitochondria, and extremely swollen astrocyte end feet. Functionally, increased BBB permeability can cause brain edema, which aggravates stroke. Therefore, we assessed BBB permeability by the Evans blue (EB) method after stroke (Fig. 4a). The stained area of the right hemisphere obviously increased in the vehicle group (Fig. 4b). The vehicle-treated group had a remarkably increased area of EB extravasation, while VK reversed the effect of MCAO-induced on BBB permeability (Fig. 4c). BBB function depends on the integrity of its components[37], mainly the microvascular endothelial cells and astrocytes involved in its formation (Fig. 4d). We confirmed that VK could maintain BBB integrity (Fig. 4e). These results collectively demonstrated that VK protects against cerebral ischemia injury by maintaining the BBB permeability after stroke in mice.

**VK reduces the secretion of proinflammatory cytokines from activated microglia.** Cerebral ischemia mediates a rapid microglial response and an abnormal activation of microglia is considered a mark of neuroinflammatory responses[38]. OGD/R treatment reduced the viability of BV2 microglia cells, which was reversed by VK treatment in a dose-dependent manner (Supplementary Fig. 4a). Additionally, OGD/R increased the expression of proinflammatory cytokines (Supplementary Table 4) in

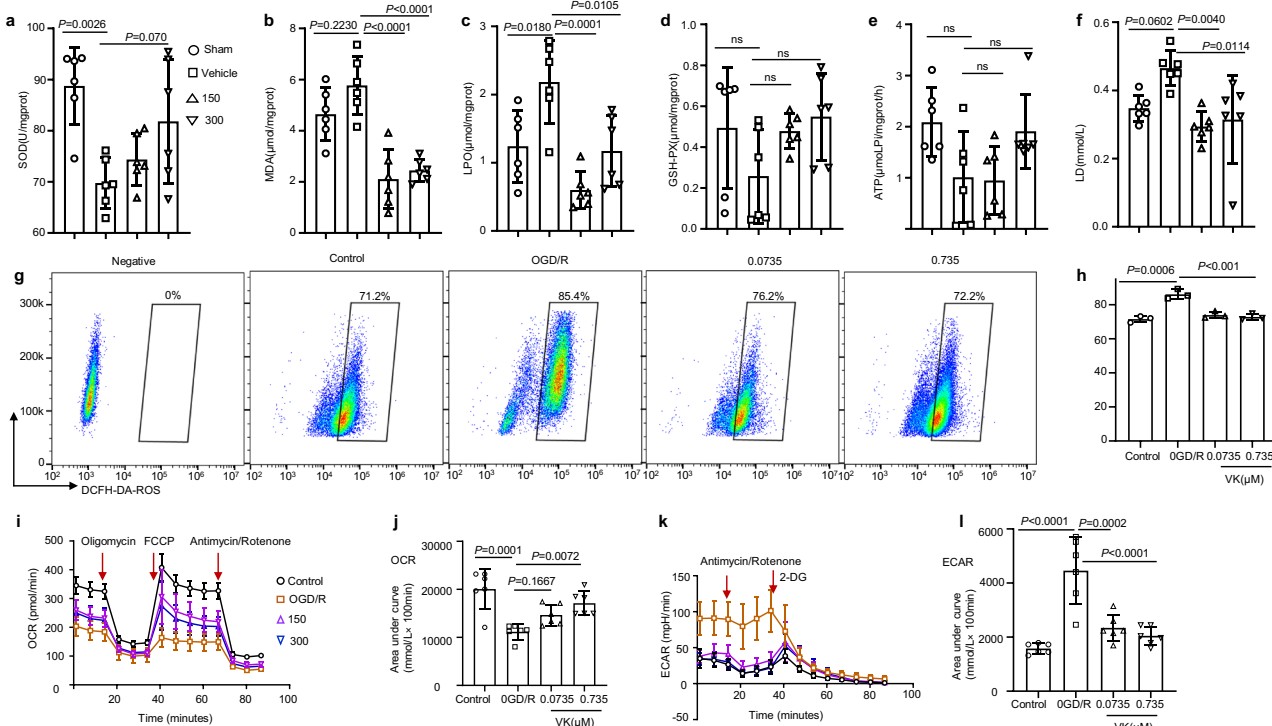

**Fig. 3 Effects of VK on oxidative stress and energy metabolism in stroke mice. a** Experimental design: in addition to the sham group, mice (8–10 weeks, male) were subjected to MCAO for 60 min. Mice were randomized into four groups: sham group, vehicle group, and VK (150 and 300 µg/kg) groups. Mice were administered VK at 0, 4, 22.5, and 46.5 h after MCAO/R as assessed by Kits. After reperfusion for 48 h, ischemic tissue was collected. **b–d** The superoxide dismutase (SOD) activity and malondialdehyde (MDA), lipid peroxide (LPO), and glutathione (GSH-PX) levels in the brain were demonstrated by the use of biochemical kits. Adenosine triphosphate (ATP synthase) (**e**) and lactic acid (LD) (**f**) were detected to examine the effect of VK treatment on energy metabolism after reperfusion injury. **g** HT22 cells stained with DCFH-DA for flow cytometry, using without DCFH-DA as a negative control for the FACS gating strategy. **h** Quantification of DCFH-DA–positive cells as the mean ± SEM from three independent experiments. DCFH-DA (10 µM). **i, j** Real-time changes in the $O_2$ consumption rate of neurons in response to treatment with the indicated concentrations of VK for 24 h. Cells were treated with 2 µM of oligomycin, 5 µM of carbonyl cyanide-ptrifluoromethoxyphenylhydrazone (FCCP), and 1 µM of rotenone and antimycin, as indicated by the three red arrows. **k, l** To assess the extracellular acidification rate, cells were treated with 1 µM of rotenone and antimycin and 50 mM of 2-deoxy-D-glucose (2-DG) as indicated by the two red arrows. Statistical analyses were performed using the Kruskal–Wallis test with the Dunn post hoc test or two-way repeated-measures ANOVA followed by Bonferroni's post hoc test. Data (animal experiment) represent the mean ± SD, $n = 5$–7. .

BV2 cells and VK remarkably blocked these effects (Supplementary Fig. 4b). The surgical operation, administration time, and detection methods were displayed in detail (Fig. 5a). Consistent with the above results, in vivo, the MCAO-induced model increased the release of the proinflammatory cytokines in the peripheral (Fig. 5b–e) and cortex of the ischemic hemisphere (Fig. 5f–i), including IL-1β, TNF-α, IL-6, and IL-8. However, in comparison with the vehicle-treated group, VK treatment decreased IL-1β, TNF-α, IL-6, and IL-8 levels in the peripheral and cortex of the ischemic hemisphere (Fig. 5b–i). Additionally, an increase in IL-10 levels was observed in VK-treated and sham mice compared to vehicle-treated mice (Fig. 5j, k). Immunohistochemical staining (IHC) was performed to observe the morphology of microglia labeled by an anti-IBA1 antibody. Cerebral I/R insult conferred a marked elevation of IBA1 expression in the cerebral cortex of mice, which was greatly blunted by the administration of VK (Fig. 5l and Supplementary Fig. 4c). From a morphological perspective, the microglia in the sham group were in a resting state characterized by small cell size and elongated branches; in the vehicle group, microglia became larger with shorter and thicker axons, and thus were in the activated state. The morphology of microglia in the VK group was between that of the previous groups. To further assess the effects of VK on activated microglia in stroke, TNF-α co-localization with IBA-1 was performed by immunofluorescence (Supplementary Fig. 4d).

We also isolated these microglia for phenotype study by flow cytometry to analyse the expression of CD45 + F4/80 + CD11b + MHCII+ (Fig. 5o and Supplementary Fig. 5). In brief, VK suppressed the excessive activation of microglia (Fig. 5m–o, Supplementary Fig. 4d, and Supplementary Fig. 6). These results suggest that VK inhibits the neuroinflammatory response and provides a new approach for the treatment of ischemic stroke.

**VK protects neurons against cell death and axonal injury in stroke mice.** In the pathophysiological process of AIS development, the destruction of neurovascular unit (NVU) homeostasis is complicated by the regulation of temporal and spatial networks during in AIS[39,40]. We explored the effects of VK on BBB damage and activated microglia. Herein, we built an OGD/R model of HT22 hippocampal neuron cells, which can remarkably decrease cell activity and cause apoptosis in HT22 cells; VK treatment enhanced cell activity (Supplementary Fig. 7a) and restrained apoptosis (Supplementary Fig. 7b, c). To further evaluate the pathological changes of neurons in stroke brains, we conducted IHC with anti-NeuN antibody and Nissl staining (Fig. 6a). The positive cells of NeuN were decreased in vehicle-treated mice when compared with the sham group. The number of NeuN-positive cells in the VK-treated group was greater than that in the

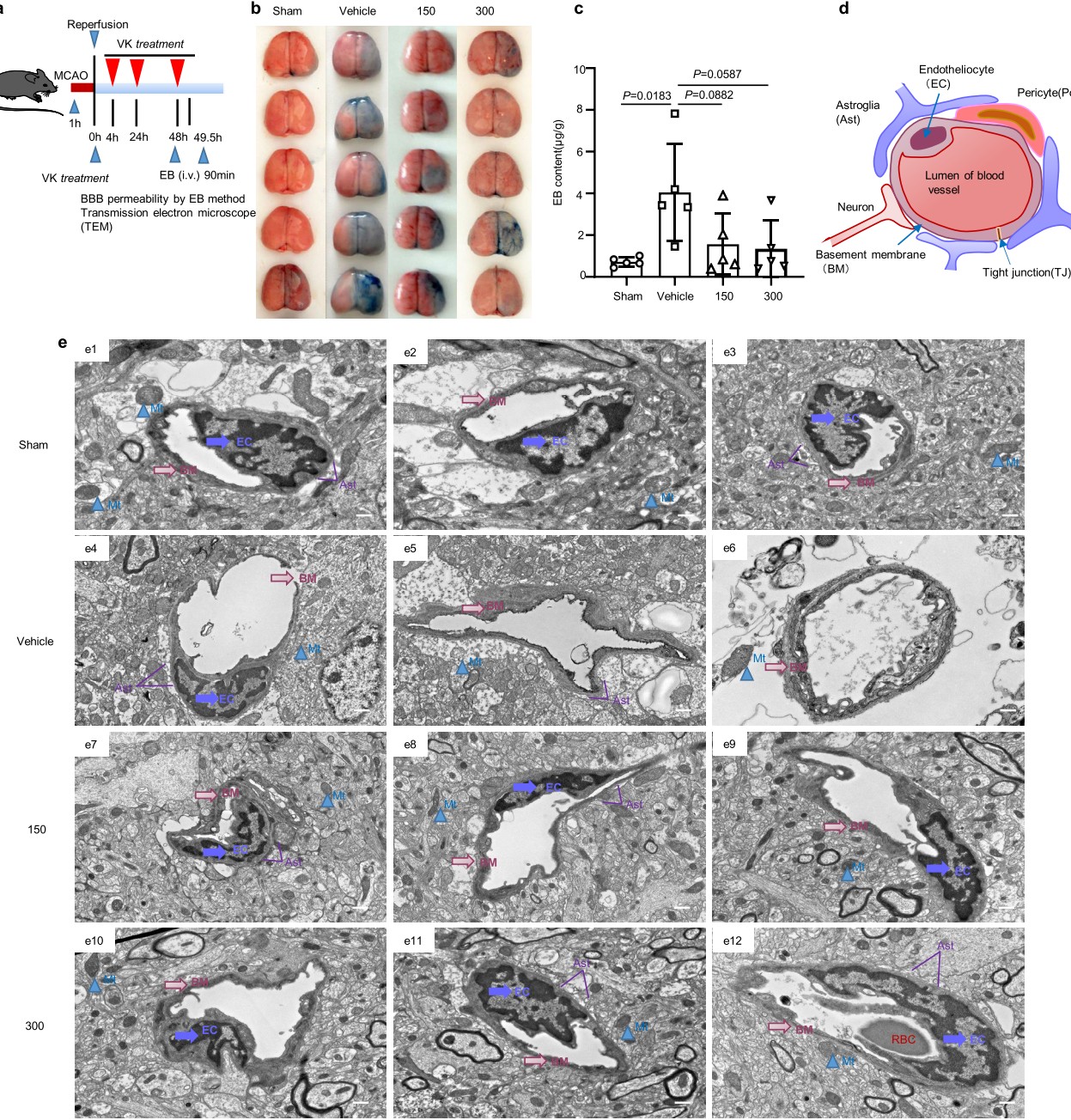

**Fig. 4 Protective effect of VK on the blood–brain barrier (BBB) in stroke mice. a** Experimental design: in addition to the sham group, mice (8–10 weeks, male) were subjected to MCAO for 60 min. Mice were randomized into four groups: sham group, vehicle group, and VK (150 and 300 µg/kg) groups. Mice were administered VK at 0, 4, 22.5, and 46.5 h after MCAO/R and BBB integrity was assessed by Evans blue (EB) and transmission electron microscope (TEM). **b** Brain samples were stained with EB. **c** EB content in damaged (right) hemisphere was quantized. **d** Schematic of BBB. The BBB mainly consists of endothelial cells (EC), pericytes (PC), astrocytes (Ast), neurons, tight junctions (TJs), and the basement membranes (BM). The main components of the BBB are cerebral microvascular ECs joined by TJs, thus restricting exogenous molecules into the brain. **e** The structure of the BBB was observed by TEM. In the sham group (**e1**–**3**), the capillary morphology was regular, the EC and TJ were complete, the thickness of the BM was uniform and continuous around the outside of EC, and the structure of mitochondria (Mt) was clear. When the BBB was destroyed (**e4**–**6**), the BM and TJ were largely dissolved and shed, and the Ast showed extreme edema; there were large electronic blank areas and the Mt was loose or vacuolar, which were relieved by VK treatment (**e7**–**12**). The marks with different symbols indicate the constituent cells or matrix of the BBB. Scale bars: 2 µm. Data are expressed as the mean ± SD, $n = 5$. Values were analyzed using one-way analysis of variance (ANOVA) with the Tukey multiple comparisons test.

vehicle-treated group (Supplementary Fig. 7d). For Nissl staining, at high magnification, typical pyknotic neurons or dark neurons were observed in selected cortex and hippocampus areas in vehicle-treated mice following stroke. Interestingly, morphologically intact neurons were broadly seen in similar brain areas of mice in the sham and VK treatment groups (Fig. 6b). The results

observably demonstrated that VK therapy inhibited cell death following ischemic damage. To confirm the functional effects of VK on neuronal activity in the peri-infarct zone, the intrinsic excitability of pyramidal neurons was recorded by electrophysiology after MCAO/R (Fig. 6c, d). Decreases in neuronal excitability could lead to the poor activation of pyramidal

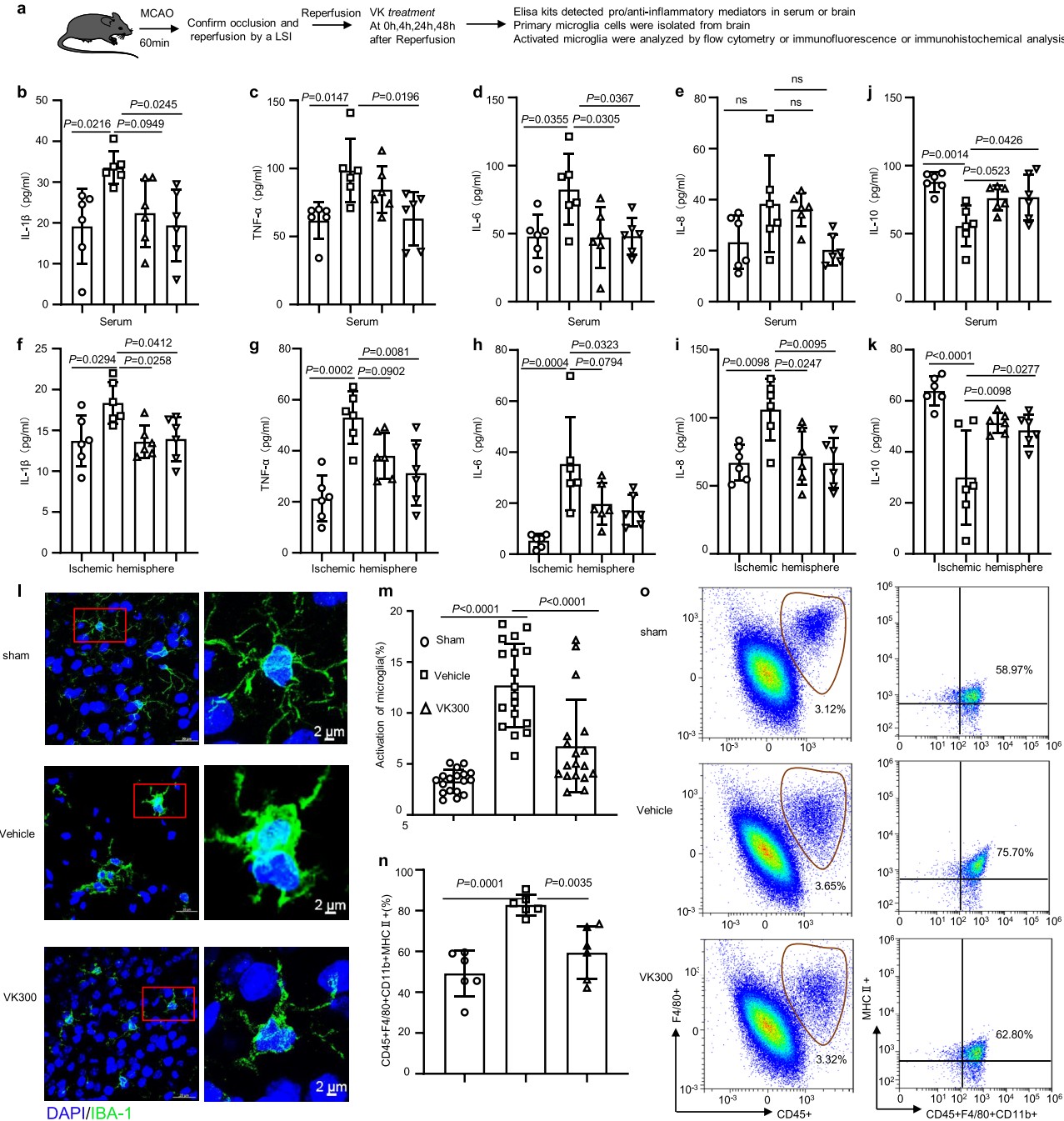

**Fig. 5 VK reduced the release of proinflammatory mediators and inhibited the activation of microglia. a** Experimental design: in addition to the sham group, mice (8–10 weeks, male) were subjected to MCAO for 60 min. Mice were randomized into four groups: sham group, vehicle group, and VK (150 and 300 μg/kg) groups. Mice were administered VK at 0, 4, 24, and 48 h after MCAO/R and the neuroinflammatory response was assessed by ELISA, flow cytometry (FC), immunofluorescence (IF), and immunohistochemical (IHC) analysis. The release of proinflammatory mediators, including IL-1β, TNF, IL-6, and IL-8, in the serum (**b–e**) or ischemic cortex (**f–i**) at 48 h after MCAO/R were detected by ELISA. **j, k** The release of IL-10 in the serum or ischemic cortex was also explored by ELISA at 48 h after MCAO/R. **l** Microglia were stained with anti-Iba-1 (green; high magnification image from a selected area, green). **m** Activated microglia were quantified. **n** Total activated microglia counts. **o** Total microglia were identified as CD45$^+$F4/80$^+$CD11b$^+$, M1-like microglia were identified as CD45$^+$F4/80$^+$CD11b$^+$ MHCII$^+$. Data are expressed as the mean ± SD, $n = 6$. Values were analyzed using one-way analysis of variance (ANOVA) with the Tukey multiple comparisons test.

neurons and a subsequent decline in the excitatory synaptic drive, thus further aggravating the ischemic pathology. Here, MCAO induction reduced the frequency of miniature excitatory post synaptic currents (mEPSCs), which is suggestive of stroke-induced excitotoxicity (Fig. 6e, f), yet showed no obvious effects on the amplitude of mEPSCs (Fig. 6g), in agreement with previous studies[41]. However, pyramidal neurons in VK-treated mice

showed a higher action potential threshold at 48 h after MCAO compared with the vehicle group (Fig. 6e). Moreover, VK also showed no obvious effects on the amplitude of mEPSCs after stroke in mice (Fig. 6f). These data indicate that VK treatment preserves the excitability of pyramidal neurons after MCAO/R in mice. Activated microglia are now generally accepted to release both protective and cytotoxic factors, through which they can

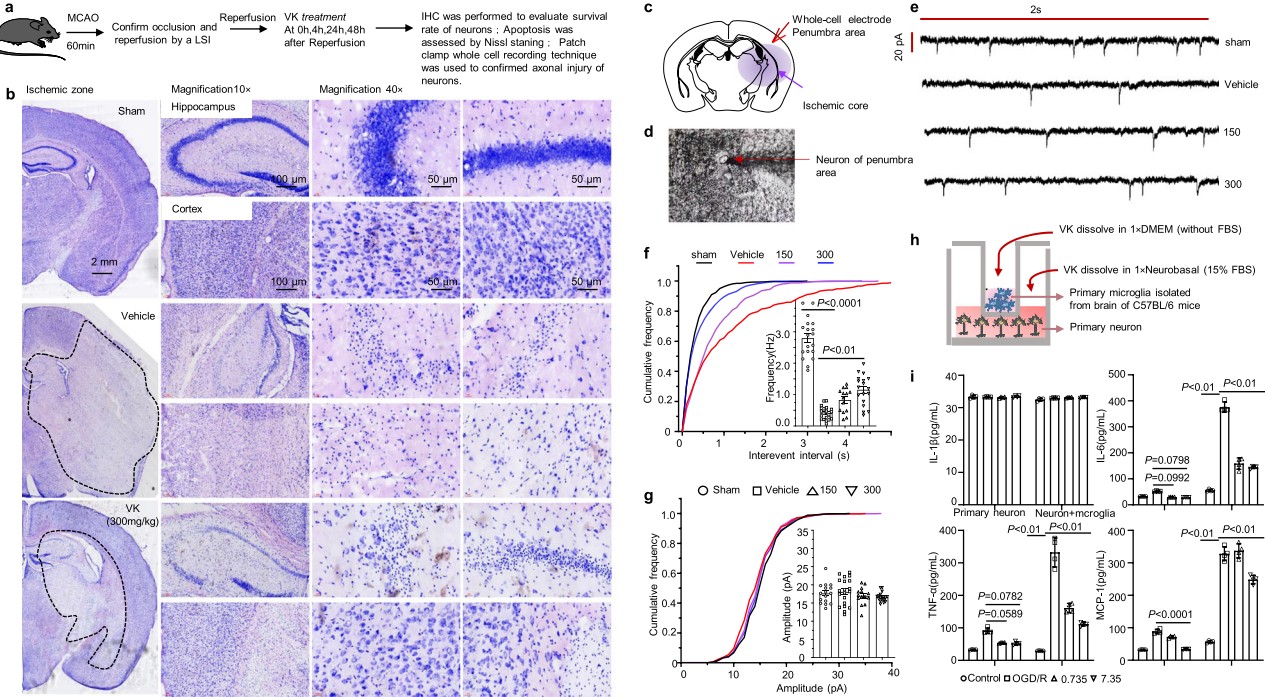

**Fig. 6 VK protects neurons against cell death and axonal injury in stroke mice. a** Experimental design: in addition to the sham group, mice (8–10 weeks, male) were subjected to MCAO for 60 min. Mice were randomized into four groups: sham group, vehicle group, VK (150 and 300 μg/kg) groups. Mice were administered VK at 0, 4, 24, and 48 h after MCAO/R as assessed cell death and axonal injury by IHC staining for NeuN⁺, Nissl staining, and patch-clamp whole-cell recording. **b** Nissl staining for the mice hippocampus (CA1 region, interaural: 2.10–2.58 mm, bregma −1.46 to −1.22 mm) and cortex in each group was displayed. Position of stimulating and recording electrodes for conduction velocity measurements in (**c**, **d**). **e** Representative firing patterns of pyramidal neurons from mouse neocortex elicited by depolarizing current steps after MCAO/R. All miniature excitatory post synaptic currents (mEPSCs) were recorded at a holding potential of −65 mV. **f** Cumulative frequency plots of the interevent interval (left) and quantitative analysis of the frequency of AMPA receptor-mediated mEPSCs (right). **g** Cumulative frequency plots of the amplitude (left) and quantitative analysis of the amplitude of AMPA receptor-mediated mEPSCs (right). Results are expressed as the mean ± SEM, $n = 5$–7. Statistical significance was determined by one-way ANOVA and Bonferroni test as post-hoc comparisons. **h** Schematic showing that primary cortical neurons obtained from fetal C57BL/6 mice of embryonic day 16–17.5 and primary microglia isolated from C57BL/6 mice at postnatal day 1–2 were co-cultured with or without VK. For the co-culture system, (**i**) cell supernatant was collected to measure proinflammatory factors. Data are mean ± SD ($n = 4$).

impact on neuronal functions and viability[42,43]. In this study, cortical neurons and microglial isolated and purified from C57bl/6 mice were co-cultured, and OGD/R models were established to investigate the protective effect of VK in vitro (Fig. 6h). The result of proinflammatory cytokines showed that VK protects neurons, and avoids persecution of proinflammatory secreted by microglia (Fig. 6i).

**Reduced neuroinflammation and apoptosis by VK treatment are associated with active PI3K–AKT and inhibitory IκBα–NF-κB signaling pathways.** To investigate the mechanisms by which VK improved functional recovery after stroke, ischemic tissue from stroke mice was collected and subjected to a phospho-antibody array. The arrays contained 17 site-specific and phospho-specific antibodies (Fig. 7a). Compared with the vehicle group, the results showed clearly decreased levels of IκBa (S32), CHK1 (S296), NF-κB (S536), p38 (T180/Y182), Casp7 (D198), JNK(T183) in the VK-treated (150 μg/kg) group, and 12 phosphorylated proteins were downregulated in the VK-treated (300 μg/kg) group (Fig. 7b, c and Supplementary Tables 5–7). Among these antibodies, the highest signal was for IκBα (S32), which was increased in MCAO-induced mice but the increase was attenuated by VK (150 and 300 μg/kg; Supplementary Tables 5–7 and Fig. 7b, c). These changes, together with the literature reports, suggest that IκBα induced the early stage of cerebral I/R injury. Similarly, VK treatment could activate the PI3K (p85α)–AKT signaling pathway and inhibit the IκBα–NF-

κB signaling pathway (Fig. 7d–h), eventually promoting functional recovery. Briefly, VK treatment stimulated MCAO-mediated PI3K (p85α) and AKT phosphorylation (Fig. 7d–f). In addition, increased phosphorylation levels of IκBα and NF-κB were seen in the vehicle-treated group compared to the sham group, while VK (300 μg/kg) reduced IκB and NF-κB levels (Fig. 7d, g, h). Additionally, VK treatment also partially down-regulated the phosphorylation levels of p38 and ERK1/2 compared to vehicle treatment (Supplementary Fig. 8). The data from Western blot analysis were consistent with a phospho-antibody array. These results suggest that PI3K–AKT signaling is activated to suppress IκBα and NF-κB levels in stroke mice by VK treatment and is related to reductions in inflammation and apoptosis.

**B2R antagonist HOE140 counteracts the neuro-protective effects of VK on stroke in mice.** B2R is neuroprotective in brain ischemic insults and increases the migration of glial cells via activated B2R[44,45]. To reveal VK-B1R or B2R interaction, the HDOCK server for integrated protein–protein docking was performed as described[46]. Known three-dimensional structure of VK and the three-dimensional structure of B1R or B2R, we predicted VK-B1R or B2R interaction sites, and the peptide docking model of VK and B2R with the highest score (Fig. 8a–d). The docking summary of the top 10 models were also displayed (Supplementary Figs. 9 and 10). Notably, VK, composed of BK and its analogs, was the first neurotoxin component isolated from wasp venom. To determine whether VK binds to B1R or B2R,

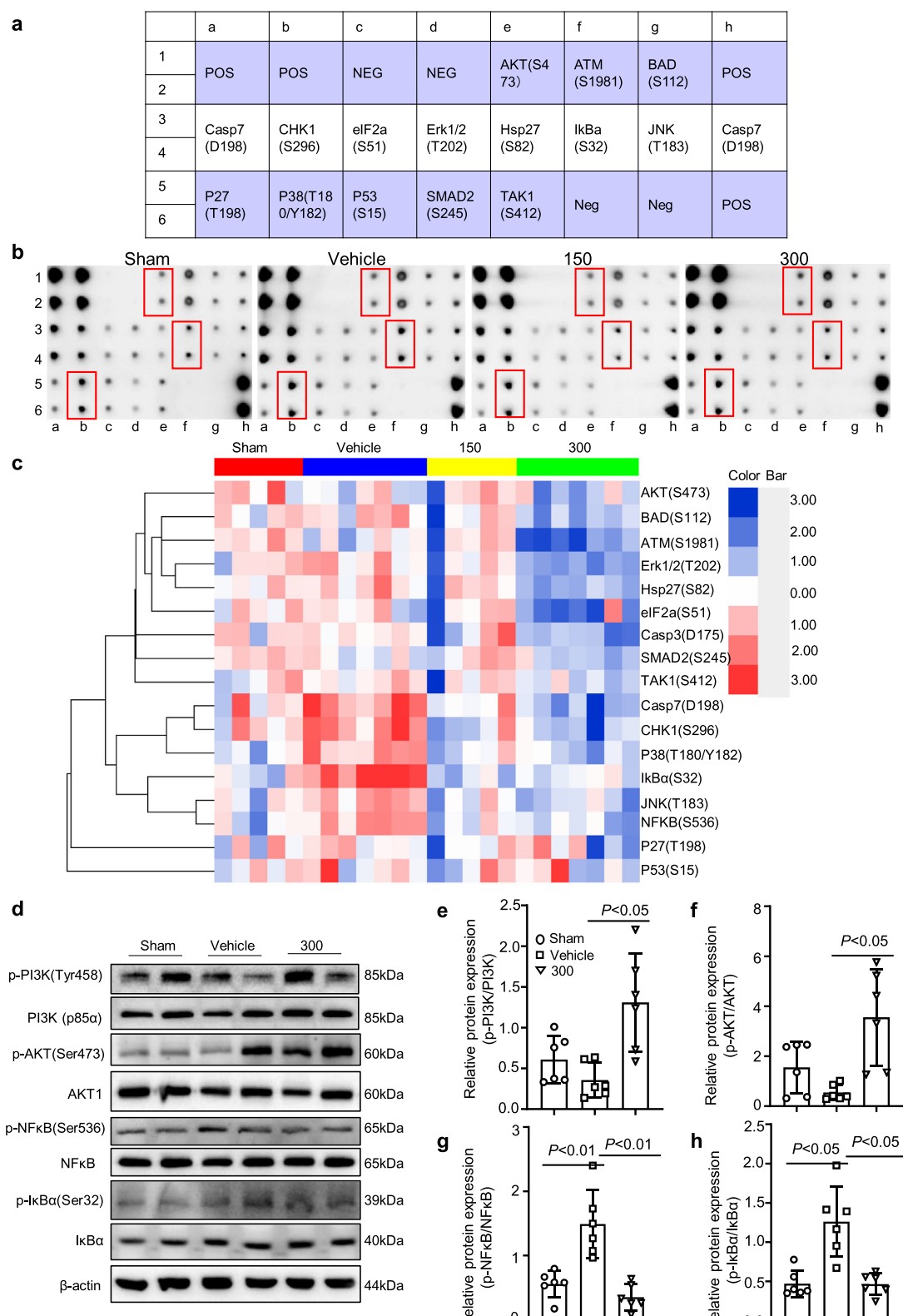

**Fig. 7 Reduced neuroinflammation and apoptosis by VK treatment were associated with PI3K–AKT-mediated NF-κB inhibition. a** Layout of antibody array. **b** Scanned image of antibody microarray and the protein expression levels were tested with antibody microarray analysis (**c**). **d–h** Western blot analysis revealed that there were changes in the phosphorylation levels of PI3K (p85α), AKT, IκBα, and NFκB in the VK groups compared to the vehicle group.

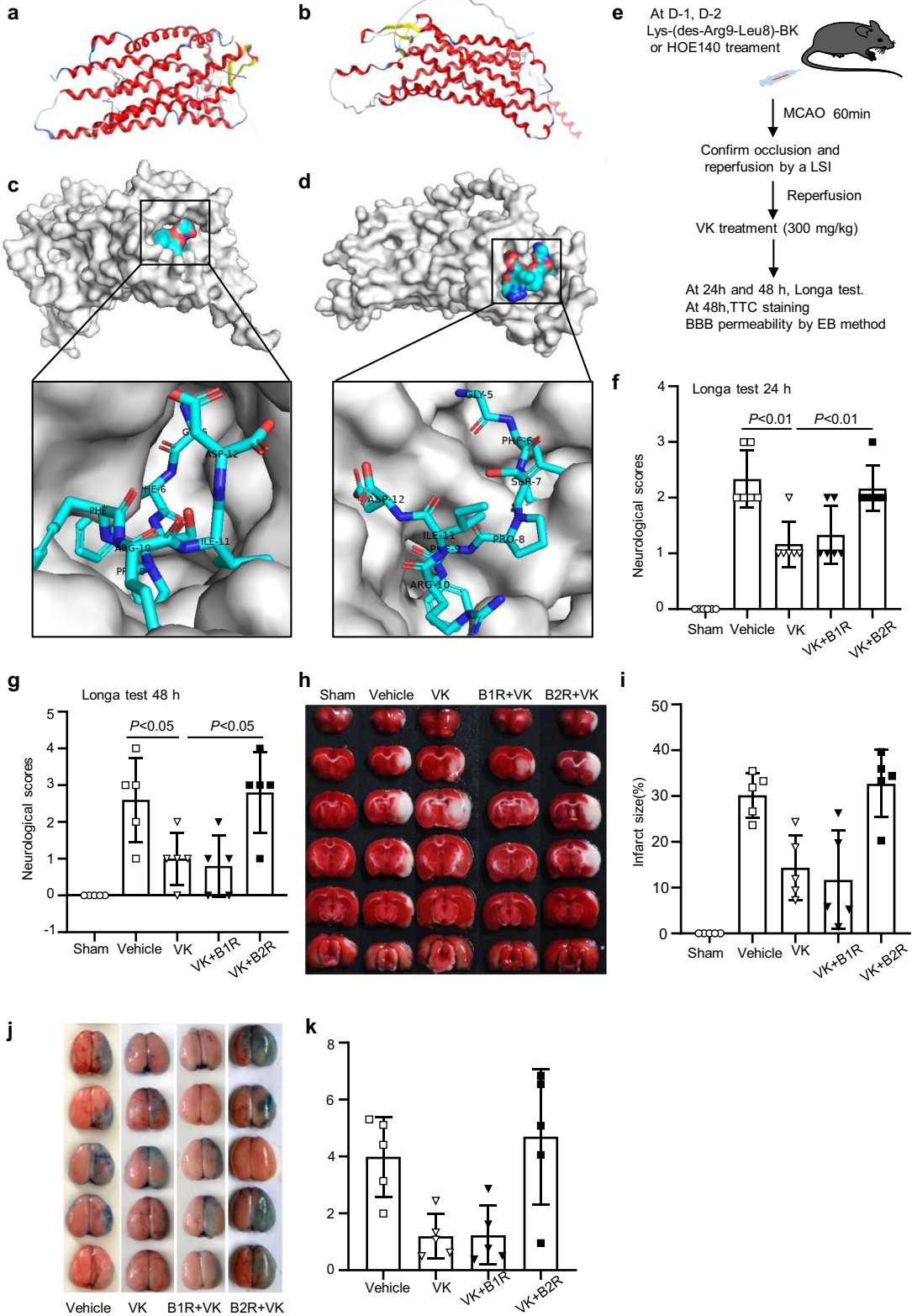

Lys-(des-Arg9-Leu8)-BK and HOE140, which are respectively inhibitors of B1R or B2R, were used to treat stroke mice (Fig. 8e). Longa test and TTC staining were conducted to assess the effects of B1R or B2R on functional recovery after MCAO/R. Compared with vehicle treatment, VK treatment attenuated neurological deficits and infarct size (Fig. 8f–i). Similarly, a combination treatment of VK and Lys-(des-Arg9-Leu8)-BK improved

functional injury. However, the major pharmacological effects of VK were blocked by HOE140 preconditioning in stroke (Fig. 8f–i). The EB extravasation assay demonstrated that the combined administration of VK and HOE140 damaged the BBB integrity, as evidenced by less EB extravasation in ischemic hemispheres (Fig. 8j, k). Mechanically, HOE140 could inhibit VK-mediated AKT activation (Supplementary Fig. 11). The B2R

**Fig. 8 VK-B1R or B2R interaction and the effect of B1R antagonist or B2R antagonist on stroke in mice. a** Secondary structure of the B1R predicted by AlphaFold (ID: AF-Q61125-F1). **b** The peptide docking model of VK and B1R with the highest score. **c** Secondary structure of the B2R predicted by AlphaFold (ID: AF-P32299-F1). α helixes, β sheets, and loops were depicted in red, yellow, and blue, respectively. **d** The peptide docking model of VK and B2R with the highest score (performed by online HDOCK SERVER: http://hdock.phys.hust.edu.cn/). **e** Experimental design: Mice (8–10 weeks, male) were randomized into five groups: sham group, vehicle group, VK (300 μg/kg) group, B1R antagonist + VK group, and B2R antagonist + VK group. The B1R + VK and B2R + VK groups were administrated Lys-(des-Arg9-Leu8)-BK and HOE140, respectively, before MCAO-induced stroke. In addition to the sham group, all mice were subjected to MCAO for 60 min. Mice were administered by VK at 0, 4, 24, and 48 h after MCAO/R and the stroke outcomes were assessed by Longa test, TTC staining, and Evans blue (EB) method. **f, g** The inhibitors affected neurological function following MCAO using the Longa test. **h** TTC staining of representative coronal sections at 48 h after reperfusion. **i** Quantitative analysis of infarct size at 48 h after MCAO/R. **j** Representative coronal brain sections showing EB extravasation. **k** EB extravasation was detected by fluorescence spectrophotometry. Data are expressed as the mean ± SD, n = 5–6. Data were analyzed using one-way analysis of variance (ANOVA) with the Tukey multiple comparisons test.

antagonist HOE140, which could suppress the upregulation of AKT signaling triggered by VK under OGD/R condition, could offset the protective effects of VK. These data suggest that targeting the VK-B2R interaction may be a practical strategy for stroke therapy.

## Discussion

Herein, VK from *Vespa magnifica* was identified by Edman sequencing, MS/MS, and amino acid sequence determination and shown to be completely degraded in plasma at 120 min, suggesting an interval time of VK administration. In this study, 150 and 300 μg/kg body weight were effective dose, these dose supports evidence from previous observations[27]. We also confirmed that safe doses of VK were <96 mg/kg in acute toxicity tests of mice, suggesting 640 times the effective dose (150 μg/kg) is still safe. Additionally, VK was shown to penetrate the BBB and live-cell imaging systems showed that VK is a cell-permeable peptide (CPP). For neuropharmacology, crossing the BBB is a remarkable challenge. The BBB is responsible for maintaining brain homeostasis via a selective permeability that protects the central nervous system (CNS). However, these characteristics also contribute to poor drug delivery and bioavailability to the CNS. Thus, crossing the BBB could increase the concentration of VK in the brain. CPPs, particularly BK, have been confirmed to selectively and transiently accelerate BBB permeability and to act as vectors for the delivery of different particles and molecules into the brain and neurons in association with compounds that have an effect on the CNS[47,48].

In the present work, BV2 microglia cells, HT22 hippocampal neuron cells, primary microglia, primary neuron as well as a mouse model of brain ischemia were utilized to investigate the effect of VK on early cerebral ischemia. Our data indicate that VK improves brain injury following ischemic stroke in mice, including the recovery of neurological impairment, the reduction of infarct volume, maintenance of the BBB integrity, and the prevention of oxidative stress, inflammatory response, and apoptosis. Stroke recovery is coordinated by a set of highly interactive processes that involve the NVU and neural stem cells[49,50]. The NVU is a structurally and functionally interdependent multi-cellular complex, comprised of endothelial cells, vascular smooth muscle cells, astrocytes, pericytes, neurons, and associated tissue matrix proteins. The NVU and combination treatment strategies for stroke have been demonstrated[51]. Emerging evidence suggests that the pathophysiological changes of AIS involve a dynamic developmental process and the pathological process involves complex temporal and spatial cascades of NVU. The changes in the early phase are characterized by a neuroinflammatory response and apoptosis. We focused on early cerebral ischemia given that excessive neuroinflammation, especially microglial activation and neuronal apoptosis, leads to the expansion of neural damage and deterioration of neurological outcomes.

The mouse MCAO model was shown to be similar to human MCAO with regards to the progression of focal neurological defects, the regional nature of the infarct, and its evolution via acute necrosis in the core, edema, inflammation, and cell death in the penumbra[52]. The induction of MCAO aggravated the severity of the CBF deficit in the first few pivotal hours during MCAO in the vehicle group, which led to a different evolution of brain injury. Hence, a successful stroke model was confirmed by LSCI. Herein, we reported that VK effectively improved the sensorimotor and cognitive recovery and reduced the infarct size in the mice model of MCAO/R. The AIS is caused by complex pathological physiological mechanisms and has different clinical manifestations in human than in mice[53,54]. The presence of standardized scores for sensory nerves and motor behavior in rats provides an advantage in functional behavioral assessment since mice and human have similar cerebral blood circulation. We demonstrated that VK improved the sensorimotor and cognitive recovery in stroke by the Longa test, grip test, rotarod test, and the Morris water maze. The potential mechanism for VK to improve functional recovery was explored. Shortly after onset of focal cerebral ischemia, neurovascular dysfunction is manifested by the disruption of BBB integrity and function. At the cellular level, endothelial cells rapidly convert into a pro-inflammatory/pro-thrombotic state via the upregulation of protease-activated receptor 1 (PAR1) and tissue factor (TF) as well as matrix metalloproteinase (MMP) gene expression in the ischemic core and boundary, which facilitates inflammation and BBB disruption. A reduction infarct size and the maintenance BBB integrity were mainly provided by VK treatment.

Furthermore, lipid peroxidation damage caused by stroke was restrained by VK treatment. Previous studies have shown that BK protects against oxidative stress-induced endothelial cell senescence[33] and BK can also be beneficial after ischemic stroke. Interestingly, VK was the first neurotoxin component isolated from wasp venom, with this small peptide playing a considerable role in regulating blood pressure, inflammation, and renal and cardiac function. We demonstrated that VK exhibited an antioxidant activity in stroke mice. The generation of reactive oxygen species is increased in activated microglia and the ensuing oxidative damage can induce an uncontrolled inflammatory condition[55]. Therefore, compounds with antioxidant activity may be beneficial for stroke and neuroinflammatory conditions. Although several studies report that BK likely causes a specific cascade of inflammatory responses in the CNS, it has also been shown to possess anti-inflammatory and neuroprotective effects, suppressing the release of inflammatory cytokines from in vitro microglia assays[25]. Similarly, our data validated that VK restrained the pro-inflammatory response and promoted the expression of anti-inflammatory mediators in OGD-induced BV2 cells and mouse primary microglia cells.

After the onset of cerebral ischemia stroke, ATP levels expeditiously decline during the first 5 min and fall to ~15–30% of the

ATP concentration measured in the non-ischemic hemisphere over the first 2 h after ischemia onset. In the present study, VK therapy partially recovered energy metabolism in the ischemic hemisphere as determined by the detection of ATP and LD content. Furthermore, to illustrate the effect of VK on energy metabolism in neurons, the glycolysis and oxidative phosphorylation pathways in neurons were also evaluated by Seahorse after stroke. Increasing evidence indicates that the neurological impairment in stroke is mediated through inflammatory reactions and pro-inflammatory cytokines. Moreover, the administration with BK after transient forebrain ischemia in rats could provide 97% neuroprotection as well as a decrease in inducible nitric oxide synthase-positive cells and caspase 3 expression and an inhibition of the release of cytosolic cytochrome. These data were in agreement with the results presented herein. In this study, we also found that treatment with VK ameliorated neuronal damage as evidenced by IHC and Nissl staining. I/R in rat brain generated a 10-fold increase in glutamate concentration during ischemia, which progressively returned to baseline after 30 min of reperfusion. This increase in extracellular glutamate levels is one of the major factors inducing cell death after brain ischemia[56]. Importantly, the neuroprotective role of VK has also been reinforced by evidence of preserving the excitability of pyramidal neurons. This finding suggests that VK is able to inhibit neuronal apoptosis and synapse injury after stroke.

The PI3K–Akt pathway has been reported to play a marked role in the neuroprotective effects against cerebral ischemia[57]. Previous studies have confirmed that PI3K–Akt signaling has anti-oxidative, anti-neuroinflammatory, and anti-apoptotic properties in neurons[58]. In contrast, the intracerebroventricular injection of LY294002, an inhibitor of AKT, reduces the phosphorylation levels of Akt and deteriorates neuronal damage after I/R. Therefore, drugs that boost AKT activity may symbolize a new class of therapeutics against AIS, which is in agreement with our experimental data demonstrating that VK could dramatically stimulate the PI3K–Akt pathway to exert neuroprotective activity. NF-κB is a central regulator of neuroinflammatory response and promotes the pro-inflammatory M1 activation of microglia. Modulating the activity of NF-κB could potentially suppress neuroinflammatory processes in ischemic stroke. In this study, we found that VK inhibited MCAO-induced phosphorylated IκBα and NF-κB p65 activation. Additionally, the protein microarray indicated that the MCAO-induced stroke model in mice could up-regulate the phosphorylation levels of IκBa (S32), NF-κB (S536), P38 (T180/Y182), Casp7 (D198), and JNK (T183), which were obviously downregulated by VK treatment. Go rich-collection analysis reveals that these processes are involved in a variety of protein functions and cell localization. However, whether the neuroprotective effect of VK is caused by active PI3K–Akt-mediated NF-κB–IκBα inhibition remains unknown.

B1R and B2R both belong to the G protein-coupled receptor (GPCR) family, with B2R mediating most of the physiological functions of kinin. It has been increasingly appreciated that BK and Lys-BK could act as a 'double-edged sword'. By inhibiting oxidative stress and apoptosis, the systemic or local delivery of human TK protects against mouse myocardial I/R injury via B2R. Similarly, we described that the neuroprotective effects of VK on stroke in mice was inhibited by HOE140, the specific antagonist of B2R. We also conducted the autodock experiment to evaluate the combination of VK and B2R. Therefore, it is presumed that the activation of VK-B2R could be protective in CNS ischemic insults. The single substitution of serine for threonine in this compound results in enhanced action when compared to BK. In terms of pharmacodynamic activity, a previous study has shown that VK exhibits remarkable anti-nociceptive effects when injected directly into the rat CNS[59]; it is approximately three times more potent and remains active longer than BK. These results can be explained by a more stable conformation in its secondary structure and/or the fact that the modification may protect against hydrolysis through neuronal kininases, preserving the effect of the peptide on B2R[59–61]. As a candidate compound acting in the CNS, the stability of VK in plasma needs to be improved by structural modification and pharmacy studies.

Finally, we only evaluated the neuroprotective effect of VK (150 and 300 μg/kg) during the period of acute cerebral ischemia in male mice. However, efficacy studies were not performed in female mice, and the protective effect of longer period time (such as 14d, 21d, 28 d) of VK on stroke severity was not demonstrated in this study. Human stroke occurs in the context of sex, aging, diabetes, hypertension, heart disease, and the use of concomitant medications, sex as a biological variable should be also considered in our future research. As far as the NVU responses are concerned, stroke is the most well-examined CNS disease because stroke pathophysiology shows relatively biphasic phenomena[62]. Under the acute phase after stroke onset, ischemic injury results in an abrupt deprivation of nutrient supplies that quickly leads to irreversible damage in the core of the affected area. On the other site, remodeling signaling such as angiogenesis and synaptic remodeling may occur in the partly preserved peri-infarct area (so-called penumbra) during the chronic phase. In future, the protective effects of VK on chronic phase of ischemic stroke will be studied and proven.

In conclusion, our data reveal that VK promotes functional recovery in mice after ischemia stroke, including an improvement of neurological impairment, a reduction of infarct volume, protection of the BBB, and the inhibition of inflammatory responses and oxidative stress. In addition, we found that reduced neuroinflammation and apoptosis by treatment with VK were associated with PI3K–AKT-mediated NF-κB inhibition. Simultaneously, the B2R antagonist HOE140 could counteract the neuro-protective effects of VK on stroke in mice. Taken together, these findings reveal that targeting the VK–B2R interaction can be considered as a practical strategy for stroke therapy (Fig. 9).

## Methods

**The purification and identification of VK**. VK was purified and identified as described previously[15]. In brief, crude venom from *Vespa magnifica* (Smith) was collected from Yunnan Province, China. The lyophilized crude venom was dissolved in deionized water (1 mg/mL) and filtered, then loaded onto a Sepax Bio-C18 column (21.2 × 250 mm, 10 μm). To purify VK, HPLC was put on a Waters 2535 system equipped with a manual injector and two-solvent system: (A) acetonitrile with 0.1% trifluoroacetic acid (TFA) and (B) water with 0.1% TFA. The effluent fractions corresponding to the chromatographic peak was manually collected in a tube and lyophilized for subsequent detection. Mass measurements of the separations were executed using a matrix-assisted laser desorption/ionization time of flight mass spectrometry (140,000 at 200 M/Z at a scan rate of 1.5 Hz, Q Exactive™, Thermofisher Scientific, USA). The molecular mass of the sample was determined by the reflection method with positive ion mode. The amino acids were confirmed by tandem mass spectrometry (MS/MS) and automatic amino acid analyzer. The purified component was sequenced with Edman degradation.

**VK stability in plasma**. VK was mixed with PBS or plasma to a final concentration of 0.735 mM and incubated (37 °C, 750 rpm; Thermomixer, Eppendorf AG, Hamburg, Germany) for 0, 30, 60, 90, or 120 min The samples were filtered with a 0.45 μm membrane and analyzed by HPLC (Agilent 1260, column: Sepax Bio C18 (4.6 × 250 mm, 5 μm); detection wavelength: 215 nm, flow rate: 1 mL/min, loading volume: 20 μL). VK in PBS or plasma was detected by quantifying the peak areas relative to the initial peak areas (0 min). All stability tests were performed at least in triplicate.

**Study approval**. All experimental procedures and animal housing in this study were designed and conducted in accordance with the approval of the Institutional Animal Care and Use Committee of Xiamen University, China (Animal Ethics no.: XMULAC20200122). Adult male C57/BL6J mice weighing 22–25 g (8–10 weeks) were obtained from Xiamen University Laboratory Animal Center. All mice were housed in a specific pathogen-free facility under a 12-h light/dark cycle in a temperature-controlled environment (22–25 °C) with a humidity of 40–70% and

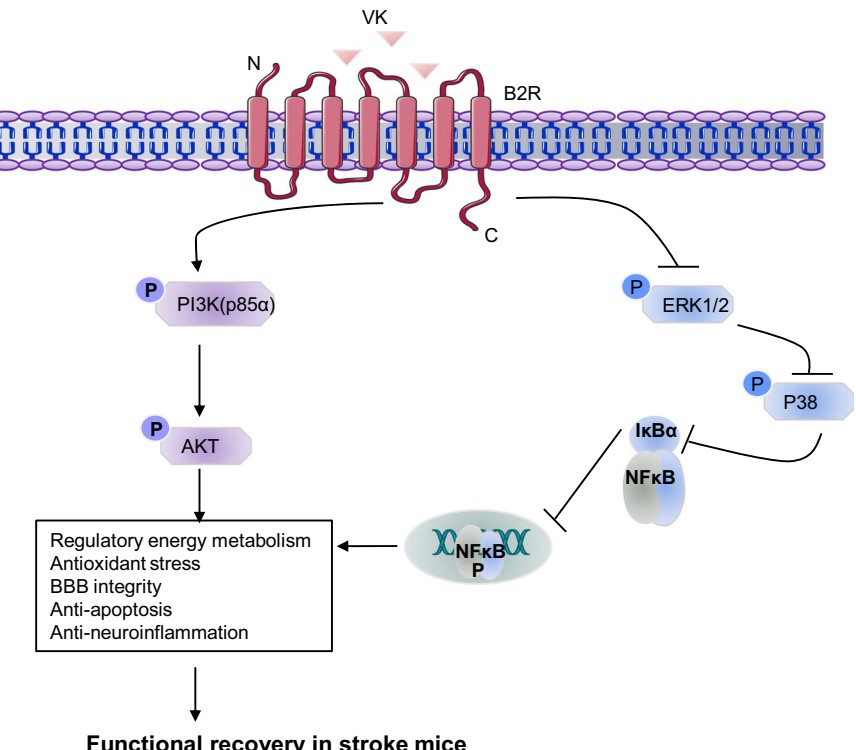

**Fig. 9 Neuroprotective effects of VK-B2R interaction in cerebral ischemia.** Targeting the VK–B2R interaction promotes functional recovery in mice after ischemia stroke, including an improvement of neurological impairment, reduction of infarct volume, maintenance of blood-brain barrier integrity, and an obstruction of the inflammatory response and oxidative stress. VK treatment led to reduced neuroinflammation and apoptosis associated with the activation of PI3K–AKT and inhibition of IκBα–NF-κB signaling pathways.

had free access to food and water. We used 357 mice for the study (Supplementary Table 1).

**Focal cerebral ischemia model.** Experimental ischemic stroke was performed by MCAO using published protocols[63,64] with some modifications. In detail, mice (8–10 weeks)[65] were deeply anesthetized with 1.5% isoflurane in a mixture of 30% $O_2$ and 69% $N_2O$. Conditions of the blood gas were maintained at constant levels throughout the operation ($P_{O_2}$, 120 ± 10 mmHg; $P_{CO_2}$, 35 ± 3 mmHg) and temperatures in the temporal muscle and rectum were also maintained at 37.5 ± 0.2 °C and 37.0 ± 0.1 °C, respectively. Body temperature was kept in the normal range (36.5–37.5 °C) with a heating lamp and a heating pad during surgery. All surgical supplies and instruments were sterilized. The internal common carotid artery (CCA) and the external carotid artery were then gently separated. The blood flow of CCA was blocked with a sterile nylon suture (No.: MSMC21B120PK50, Ruiwode Bio-technology Co., Ltd., China), which was inserted into middle cerebral artery from CCA. The mice in the sham group underwent the same surgery without a nylon suture. After 60 min, the suture was removed and the blood circulation was recovered, MCAO/R was induced. Regional CBF was monitored in all stroke animals by LSCI (PeriScan PSI System; PERIMED, Stockholm, Sweden) to confirm a successful occlusion.

**Exclusion criteria.** Exclusion criteria in mice were performed as described above:[65] (1) mice with a CBF reduction of <70% were excluded from further assessments; (2) no neurological deficits at 3 h after MCAO/R; (3) postmortem examination shows subarachnoid hemorrhage.

**Drug treatment.** Mice were randomly divided into four groups: (1) the sham group, (2) MCAO group (vehicle treated), and test drug groups (3) MCAO + VK (150 μg/kg, i.p.), and (4) MCAO + VK (300 μg/kg, i.p.). After MCAO, mice were treated with normal saline or VK 0, 4, 22.5, and 46.5 h.

**Behavior test.** Researchers who were blinded performed all behavioral tests.

*Longa test.* Mice were graded on a 5-point scale, which was modified from the Longa score.

*Grip test.* The grip strength (The Chatillon® DFE Digital Force Gauge DFX-050) was used to evaluate the effects of VK on muscular incoordination of mice.

The forelimbs of the mice were placed on the test grid and the mice were gently pulled after the animals were grasped and the force value when the claws of the animals were recorded was the grip value. The test was performed at 24 and 48 h after MCAO/R.

*Rotarod test.* The mice were placed in a RotaRod Treadmill (Harvard Apparatus, LE8205) for adaptive training 3 days before MACO/R. The rotating rod was started and the speed was set to accelerate to 40 rpm within 5 min. The mice drop time was recorded as an indicator. The test was performed at 24 and 48 h after MCAO/R.

*Morris water maze test.* To evaluate the spatial learning and memory ability of mice, the Morris water maze was performed after MCAO/R at 11, 12, 13, 14, and 15 days as described previously with minor modifications[66,67]. Briefly, the Morris water maze consists of a black circular pool ((109 cm diameter), a constant temperature system (24 ± 1 °C), a circular platform (11 cm diameter), direction markers, flood-light curtains, and a camera system. The test was divided into two parts: the place navigation test (learning) and the probe test (memory). The purpose of the place navigation test is to evaluate the capacity of the mice to detect the submerged platform by memorizing external spatial cues. Each mouse was placed in one of four quadrants in the water and was allowed 60 s to find the platform during the test. To promote mouse memory of the external spatial cues displayed around the room, it was placed on the platform for another 30 s after each trial ended. Three trials were performed per day with the platform present and continuously detected for 4 days. Data from those trials were shown as the latency to locate the hidden platform on each day and were in indirect proportion to spatial learning abilities. A single 60 s probe test was performed on each mouse on the final day of testing. The platform was withdrawn and each mouse was placed at the same starting position in the pool and allowed to swim for 60 s. The investigator then recorded the time spent in the target quadrant, where the platform had previously been located. The latter measurements were interpreted as spatial memory.

**Morphological assessment**

*TTC staining.* 2,3,5-triphenyltetrazolium chloride (TTC) staining was used to assess cerebral infarction. The TTC solution was prepared at 37 °C in the dark with phosphate buffer (0.2 M $Na_2HPO_4$ and 0.2 M $NaH_2PO_4$, pH 7.4) immediately before use. For this assessment, the brains were rapidly isolated and sliced into 2-mm-thick coronal sections. The brain slices were then stained in 0.2% TTC solution at 37 °C for 15 min and then fixed with 4% paraformaldehyde for 24 h. Infarct size was analyzed using Image Pro Plus 6.0 image analysis software.

**Nissl staining**. Mice were anesthetized at 48 h after MCAO/R. Brain samples (bregma −2.06–1.94 mm) were collected and fixed by 4% paraformaldehyde for 12–18 h. Following sequential dehydration by 10%, 20%, and 30% sucrose, brains were put in frozen section embedding medium (SAKURA Tissue-Tek O.C.T. Compound, Japan) and cut into 25-mm-thick sections at the level of the frontal cortex (bregma: 1.0 to −1.94 mm) using a cryostat microtome (Leica CM1950). The sections were subsequently stained with Nissl staining.

**MRI**. After MCAO/R, mice were treated with VK (300 μg/mL, i.p.) at 0, 4, 22.5, and 46.5 h. Mice were deeply anesthetized with 1.5% isoflurane in a mixture of 30% $O_2$ and 69% $N_2O$. The magnetic resonance imaging (MRI) experiments were performed on a horizontal bore 9.4 Tesla scanner operating on a Bruker Avance platform (Bruker 9.4 T Biospec). T2-weighted imaging was used to evaluate the infarct volume. Three mutually perpendicular images were obtained to localize the infarction site. Then, 13–15 T2-weighted axial slices (1 mm thick) covering the entire damage area (field of view = 30 × 30 mm, matrix size = 256 × 256, echo time[3] = 33 ms, repetition time[68] = 3500 ms) were acquired by a spin-echo pulse sequence to delineate anatomical details and calculate qualitative T2 maps[69].

**Electrophysiological recordings and assessment of excitatory inputs**. Mice were treated with VK (150 and 300 μg/mL, i.p.) at 0, 4, 22.5, and 46.5 h after MCAO/R. Pyramidal neurons in layers 2–3 of the mouse neocortex located ~1 m$^2$ were prepared as previously described[41,70]. Following incubation, slices were transferred to a recording chamber in which oxygenated artificial cerebrospinal fluid was warmed to 32 °C and superfused over the submerged slices at 2 mL/min. Recordings were collected from the pyramidal neurons, with mEPSCs recorded at a holding potential of −70 mV in the presence of tetrodotoxin to inhibit action potential-mediated excitatory postsynaptic currents. For mEPSC recording, the composition of artificial cerebrospinal fluid (in mM) was as follows: 120 K-gluconate, 20 KCl, 10 HEPES, 2 $MgCl_2$, 0.1 EGTA, 10 sodium phosphocreatine, 0.2 leupeptin, 4 Mg-ATP, 0.3Na-GTP, and pH 7.3 (290 mOsm). mEPSCs were recorded at a holding potential of spontaneous and miniature events were analyzed by using the MiniAnalysis program (Synaptosoft, Inc.).

**Immunofluorescence staining**. The sections were stained with immunofluorescence as previously described[71]. The antibodies used and their concentrations are listed below (antibody, dilution, catalog number, company, and country): anti-IBA1 (1:400, 019-19741, WAKO, Japan) and anti-NeuN (1:400, ab177487, Abcam, USA). After primary antibody incubation, sections were washed four times with PBS for 10 min at room temperature and incubated with the secondary antibody (Goat anti-Rabbit IgG (H + L) Cross-Adsorbed Secondary Antibody, Alexa Fluor 488, Thermo) for 1 h. For negative control staining, the primary antibody was omitted during immunostaining. Images were acquired by confocal microscopy (Zeiss LSM 880 or FV1000 MPE-B, Olympus) and processed with ImageJ (NIH, Bethesda, MD, USA).

**Antibody microarray analysis**. Protein phosphorylation detection was performed with a Phospho Explorer Antibody Array (Catalog number AAM-APOSIG-1-8, Ray Biotech, USA), allowing for the examination of 17 phosphorylation sites (three replicates per antibody). Whole-cell lysates from the prefrontal cortex in mice after MCAO/R and VK-treated, and the assay was performed according to the manufacturer's protocol.

**Quantitative immunoblotting**. Quantitative immunoblotting was accomplished as previously described[72]. Anti-β-actin (Themo) was used as an internal loading control. Imaging of blots was performed using an AzureC300 with secondary antibodies(Thermo). Analysis was carried out in Image Lab and signal intensities were normalized to loading controls, where applicable. The antibodies and their concentrations are listed as follows (antibody, dilution, catalog number, company, country): AKT (1:1000,4691, CST, USA), phospho-AKT (1:1000, 4046S, CST), phospho-p44/42 MAPK (Erk1/2) (Thr202/Tyr204) (1:1000, 4370S, CST), p44/42 MAPK (Erk1/2) (1:1000,4695S, CST), p38 MAPK (1:1000,8690S, CST), phospho-p38 MAPK (Thr180/Tyr182) (1:1000, 4511S, CST), NF-κB p65 (1:1000,8242S, CST), phospho-NF-κB p65 (Ser536) (1:1000,3033S, CST), IκBα (L35A5) Mouse mAb (1:1000, 4814S, CST), phospho-IκBα (Ser32) (1:1000,2859S, CST), PI3 kinase p85α (1:1000,11889S, CST), and phospho-PI3 kinase p85 (Tyr458)/p55 (Tyr199) (1:1000, 17366S, CST).

**Transmission electron microscopy**. Frontal cortex samples were isolated from mice with stroke and fixed in 2% glutaraldehyde and 4% paraformaldehyde in 0.1 M sodium cacodylate (pH 7.4), treated with 10% gelatin solution in sodium cacodylate buffer, and incubated with 2% osmium tetroxide. Sections were cut using a Leica EM TP ultramicrotome (Leica Microsystems) at 40 nm and placed within grids stained with a 1:1 mix of 3% uranyl acetate and 50% acetone for 30–60 s. Grids were imaged with a JEM 1400 transmission electron microscope (JEOL) at ×1200 for low magnification and ×12,000 for high magnification (unless otherwise noted) using Gatan Microscopy Suite software (Gatan).

**Primary microglia and flow cytometry**. Brain samples were collected in sham or MCAO mice, digested with collagenase IV and DNase, and filtered with sterile cell strainers (Biologix, 70 μm). Single-cell suspensions were prepared after the removal of red blood cells by ammonium-chloride-potassium (ACK) lysis buffer and isolated leukocytes in brains by 30%, 37%, and 70% Percoll (GE Healthcare) density gradient centrifugation, respectively. Cell numbers were counted by a Coulter counter (Thermo Fisher). Cells were washed with buffer (PBS with 0.5% bovine serum albumin and 0.02% sodium azide) three times and subsequently stained with fluorochrome-conjugated monoclonal antibodies: PE Rat anti-Mouse CD45 (553081, BD Biosciences), FITC anti-mouse F4/80 antibody (123108, Biolegend), PerCP-Cyanine5.5 CD11b antibody (45-0112-82, Invitrogen), and APC anti-mouse I-A/I-E antibody (107614, Biolegend). Samples were analyzed using Flow Cytometry (CytoFLEX S). Subsequent analysis was performed with FlowJo software (Tree Star Inc., San Carlos, CA, USA). Total microglia were identified as CD45$^+$F4/80$^+$CD11b$^+$, M1-like microglia were identified as CD45 + F4/80 + CD11b + MHCII + .

**Biochemical assays**. Mice were sacrificed under ether anesthesia 3 days after stroke. The brain was collected, weighed, and placed in a homogenizer for homogenization. Using 0.9% normal saline (NS) as a homogenization medium, a 10% tissue homogenate was prepared, centrifuged at 3500 rpm for 10 min, and the supernatant was taken. Using 0.9% NS as a homogenization medium, a 10% brain tissue homogenate was prepared, centrifuged at 3500 rpm for 10 min, and the supernatant was extracted. Kits for ATP, glutamic acid, glutathione, lactic acid, LPO, MDA, and SOD were purchased from Nanjingjiancheng Bioengineering Institute, and were measured as previously described for specification.

**Cytokine enzyme-linked immune sorbent assays**. Mice were decapitated under anesthesia at 48 h after MCAO/R. The IL-1β (Cat# 88-5019-88), TNF-α (Cat# 88-7324-22), IL-6 (Cat# 88-7064-88), and IL-8 (Cat# 88-8086-22) were purchased from Thermo Fisher Scientific. The levels of these proteins were measured by enzyme-linked immune sorbent assays (ELISA) kits, according to the manufacturer's recommendations.

**OCR and ECAR**. HT22 cells were counted and plated at 8000 cells per well in a Seahorse XF96 Cell Culture Microplate for all experiments (Agilent). They were then treated with VK (0.0735 or 0.735 μM) for 24 h. After 24 h of incubation, the OGD/R model was established as described previously and established with Agilent Seahorse XF medium (Agilent) supplemented (1 mM pyruvate, 2 mM L-glutamine, and 10 mM D-glucose, respectively; 525 μL was injected into each well). Next, HT22 cells were washed after 24 h of incubation and were incubated in a 0% $CO_2$ chamber at 37 °C for 1 h, before being placed into a Seahorse XFe96 Analyzer (Agilent). In OCR experiment, HT22 cells were treated, with 2 μM of oligomycin, 5 μM of FCCP, and 1 μM of rotenone and antimycin; to assess ECAR, they were treated with 1 μM of rotenone and antimycin and 50 mM of 2-DG. All OCR and ECAR data were normalized to cell number per well using WAVE (Agilent).

**FITC-labeled VK and live-cell imaging**. VK (Gly–Arg–Pro–Hyp–Gly–Phe–Ser–Pro–Phe–Arg–Ile–Asp–NH2) was labeled with FITC in the N-terminus. VK (100 nM) was added to each well and, after 15 min of incubation at 37 °C, live-cell images were acquired and processed by confocal microscopy (Zeiss LSM 880). Live-cell images were recorded for 90 min. The negative group was FITC treated without labeled VK.

**Fluorescence imaging of VK**. A successful MCAO model was established and these mice were used for subsequent experiments. Immediately after surgery, FITC or FITC-labeled VK were administered intravenously and mice were killed after 30 min to isolate the brain and imaged by an imaging system (IVIS Lumina II). For fluorescent imaging, the brain was immediately collected and kept in the dark for 0, 30, 60, and 90 min after the administration of VK. Frozen sections were prepared according to the above method (seen Morphological Assessment). Images were acquired by confocal microscopy (FV1000 MPE-B, Olympus).

**The HDOCK server for integrated protein–protein docking**. Protein–protein and protein-DNA/RNA docking based on a hybrid algorithm of template-based modeling and ab initio free docking as consulted http://hdock.phys. hust.edu.cn/ [46].

**Statistical analysis**. Statistical analysis was conducted using Graph Pad Prism 8 software. In order to determine the normal distribution of the sample, the Kolmogorov–Smirnov test was conducted. If the sample was normal distribution, then statistical analysis was conducted among multiple groups using one-way analysis of variance (one-way ANOVA), followed by the Dunnett test or Student's $t$ test. If the sample was not normal distribution, then a Kruskal–Wallis test was performed. A statistical difference was established at $P < 0.05$.

**Reporting summary**. Further information on research design is available in the Nature Research Reporting Summary linked to this article.

## Data availability

All data is included in the article and supplementary information. Source data of figures are provided in Supplementary Data 1. The protein microarray and uncropped western blots are provided in Supplementary Figs. 12–19. The data that support the findings of this study are available from the corresponding author upon reasonable request.

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

## Acknowledgements

The experiments were mainly carried out in the Center Laboratory, Xiang'an Hospital of Xiamen University. The authors would like to thank Core Facility of Biomedical, Xiamen University for assistance with experimental apparatus. The authors would like to also thank Core Facility of Biomedical, Xiamen University (Xiang You, Jingru Huang, and Haiping Zheng). This research was supported in part by National Natural Science Foundation of China [grant number: 82106822, 81903924, 81703742, and 81360679]; The Natural Science Foundation of Yunnan Province [grant number: 2017FA050 and 2019FB121]; The Special Program of Science and Technology of Yunnan Province (202002AA100007).

## Author contributions

Z.H.R. designed the project and wrote the manuscript. Z.H.R., W.M., G.Y., H.F.R., L.J.M., X.D., W.Q. and L.W.D. constructed stroke mice. W.M., G.Y. and Z.H.R. performed behavioral tests, Immunofluorescence, Immunostaining, Western blot, Transmission electron microscope, and data analysis. L.J.G. performed four coagulation tests. Z.H.R. performed Flow cytometry. W.X.M. and X.H. performed platelet function tests. Z.H.R. and W.X.M. performed data analysis of antibody arrays. Z.H.R. and W.M. performed primary neuron and microglia culture. C.J.D., Z.C.G. and Z.Y. helped with data analysis, interpretation, and supervised the project. Z.C.G., C.J.D. and Z.Y. were responsible for review and editing. Z.H.Y. provided the source and identification of VK.

## Competing interests

The authors declare no competing interests.
