## [Transparent Peer Review File · Communications Biology]

Reviewers' comments:

Reviewer #1 (Remarks to the Author):

The manuscript entitled "Vespakinin-M, a natural peptide from *Vespa magnifica*, promotes functional recovery in stroke mice" by Dr. Zhao et al. describes that vespakinin-M (VK), a natural peptide from wasp venom, promotes functional recovery in mice after ischemia stroke, including an improvement of neurological impairment, a reduction of infarct volume, protection of the BBB, and the inhibition of inflammatory responses and oxidative stress. Mechanistically, they found that reduced neuroinflammation and apoptosis by treatment with VK were associated with PI3K-AKT-mediated NF- κ B inhibition. The results are potentially interesting and the techniques used in this study is appropriate to conclude the results. However, there are some major issues that should be addressed:

1. How was the dose (150 and 300 μ g/kg) determined? any references? Will increasing the dose of VK yield better protection effect in vivo?
2. Oxidative stress is caused by elevated production of reactive oxygen species (ROS). It can cause neuron apoptosis. The level of ROS should be detected in figure 3.
3. In this study, microglia and neuron were used to do some experiments in vitro. From my point of view, the experiments of OGD induced co-culture of these two kinds of cells should be done. Because VK has effect on both microglia and neuron, and in vitro co-culture experiment can really reflect the situation in vivo.
4. Please note that Iba1 is not a specific marker for microglia; it can also be used for macrophages. Therefore, the results from figure 5 did not necessarily mean microglia activation.
5. The authors did not provide exclusion criteria for the MCAO model. Are all the mice included in this study? And the authors did not show the mortality rate of the animals. Did all the mice survive? Please provide the data.
6. There is no information about mice in the method part, and why did the authors use 3-month year old mice? Many studies used 2-month year old mice.
7. Please explain that "brain tissues" in this study stand for which part of the brain?
8. I would like to suggest the authors to conduct the autodock experiment to evaluate the combination of VK and B2R or show us some other direct evidence to prove your conclusion.
9. Figure 1e lacks scale bar and the abscissa of Figure 7g is wrong. Many symbols are missing in the current manuscript. Please check the whole manuscript carefully during the revision process.

Reviewer #2 (Remarks to the Author):

In the current manuscript Zhao H et al investigated the role of Vespakinin-M, a natural peptide from *Vespa magnifica*, on infarct volume and post-stroke functional recovery in preclinical stroke mouse model. The authors showed 1) the structure of VK by HPLC and stability/distribution test; 2) effect of VK on infarct volume, functional outcome post-stroke; oxidative stress; inflammation; BBB integrity and cell death and axonal injury; 3) for mechanism of VK on neuroprotection, the authors found VK treatment is associated with activated PI3K-AKT pathway and NF- κ B pathway inhibition.

This is a very well performed study. Overall, this reviewer was impressed with the body of work as presented, and it is certainly of broad significance across neuroscience fields. The experiments as presented are elegantly designed and straightforward in their interpretation towards defining a role for VK on stroke severity. This reviewer has a few minor concerns and clarifications that would benefit this manuscript prior to publication.

1. RIGOR guidelines should be followed for all effective translational stroke research. Specifically for this manuscript, female mice and a second stroke model should be used to investigate whether VK is only effective in male mice or MCAO model; longer period time of stroke severity should be demonstrated.
2. Systemic toxicity test for VK, such as liver/ kidney, and especial those related to coagulation, platelet function and bleeding should be performed.

3. The author do not demonstrated whether VK binds to B1R orB2R, the use of a single antagonist with one in vivo study does not reveal VK-B2R interaction, nor a practical strategy for stoke therapy- such a statement is not accepted.

4. In Fig. 7c lack of legend. Please check typos throughout the manuscript. For example, line 916 and 972, "1" for "i" , et al.

Reviewer #1 (Remarks to the Author):

The manuscript entitled "Vespakinin-M, a natural peptide from *Vespa magnifica*, promotes functional recovery in stroke mice" by Dr. Zhao et al. describes that vespakinin-M (VK), a natural peptide from wasp venom, promotes functional recovery in mice after ischemia stroke, including an improvement of neurological impairment, a reduction of infarct volume, protection of the BBB, and the inhibition of inflammatory responses and oxidative stress. Mechanistically, they found that reduced neuroinflammation and apoptosis by treatment with VK were associated with PI3K-AKT-mediated NF- κ B inhibition. The results are potentially interesting and the techniques used in this study is appropriate to conclude the results. However, there are some major issues that should be addressed.

Thank you very much . We appreciate your constructive comments and suggestions on our manuscript entitled “Vespakinin-M, a natural peptide from *Vespa magnifica*, promotes functional recovery in stroke mice”.

1. How was the dose (150 and 300 μ g/kg) determined? any references? Will increasing the dose of VK yield better protection effect in *vivo*?

Determined the dose (150 and 300 μ g/kg) consistent with effective dose of bradykinin1. We have added reference Line 335-338 (discussion). Ischemic stroke were treated VK (75, 150, 300 μ g/kg) in preliminary experiment. Infarct size and behavior were evaluated. Compared with MCAO/R group, decreased area of cerebral infarction and mortality were seen in VK group. However, there were no significant differences between the three groups (75, 150, 300 μ g/kg). The effect of increased the dose of VK was not illustrated in this paper. But, we are working on the long-term pharmacodynamics of VK on ischemic stroke, such as angiogenesis and synaptic remodeling.

2. Oxidative stress is caused by elevated production of reactive oxygen species (ROS). It can cause neuron apoptosis. The level of ROS should be detected in figure 3.

Thank you. ROS level have be demonstrated using a DCFHDA kit. We have added to Figure 3 .Lines 184-188.

3. In this study, microglia and neuron were used to do some experiments in vitro. From my point of view, the experiments of OGD induced co-culture of these two kinds of cells should be done. Because VK has effect on both microglia and neuron, and in vitro co-culture experiment can really reflect the situation in *vivo*.

Thanks. Our experimental design is lacking in this part. For co-culture experiment, primary microglia and neuron extracted from newborn or embryonic mice. We confirmed VK protects neurons, and avoids persecution of proinflammatory secreted by microglia. Figure 6 and lines 276-282.

4. Please note that Iba1 is not a specific marker for microglia; it can also be used for macrophages. Therefore, the results from figure 5 did not necessarily mean microglia activation.

Thank you very much. Iba1 is a marker for microglia, activated microglia can be defined as larger with shorter and thicker axons from morphology. As you say, it's not objective. Therefore, TNF- α co-localization with IBA-1 was performed by immunofluorescence (Supplementary Fig. 4d). We also isolated these microglia for phenotype study by flow cytometry to analyse the expression of CD45+F4/80+CD11b+MHCII+(M1 polarization). Lines 239-244.

5. The authors did not provide exclusion criteria for the MCAO model. Are all the mice included in this study? And the authors did not show the mortality rate of the animals. Did all the mice survive? Please provide the data.

Exclusion criteria have been added. Lines 517-521.

All sham-operated mice survived, but the overall mortality of MCAO was 29.06% (Supplementary Table.1 and Table.2). Lines 85-89. Thank you.

6. There is no information about mice in the method part, and why did the authors use 3-month year old mice? Many studies used 2-month year old mice.

Thank you. In fact, 8-10 weeks male mice were used to build MCAO/R, we have revised and added reference. Line 496 and line 503.

7. Please explain that “brain tissues” in this study stand for which part of the brain?

For the MCAO model, the infarct size implicated caudate-putamen, striatum, and cortex. This study focus on the cortex of ischemic hemisphere. Thank you for your suggestion, we have revised, Such as Line 226-229.

For rozen section, brain (interaural 1.7-4.30 mm, bregma -2.06-1.94 mm) was collected Lines 594.

8. I would like to suggest the authors to conduct the autodock experiment to evaluate the combination of VK and B2R or show us some other direct evidence to prove your conclusion.

According to your suggestion, To reveal VK-B1R or B2R interaction, the HDOCK server for integrated protein-protein docking was performed. we predicted VK-B1R or B2R interaction sites (Fig.8 a-d) , and the peptide docking model of VK and B1R with the highest score. The docking summary of the top 10 models were also displayed (Supplementary Fig.7 and Fig.8). Lines 310-315.

9. Figure 1e lacks scale bar and the abscissa of Figure 7g is wrong. Many symbols are missing in the current manuscript. Please check the whole manuscript carefully during the revision process.

Thank you. We have carefully checked and revised.

1. Danielisova V, *et al.* Bradykinin postconditioning protects pyramidal CA1 neurons against delayed neuronal death in rat hippocampus. *Cell Mol Neurobiol* **29**, 871-878 (2009).

Reviewer #2 (Remarks to the Author):

In the current manuscript Zhao H et al investigated the role of Vespakinin-M, a natural peptide from *Vespa magnifica*, on infarct volume and post-stroke functional recovery in preclinical stroke mouse model. The authors showed 1) the structure of VK by HPLC and stability/distribution test; 2) effect of VK on infarct volume, functional outcome post-stroke; oxidative stress; inflammation; BBB integrity and cell death and axonal injury; 3) for mechanism of VK on neuroprotection, the authors found VK treatment is associated with activated PI3K-AKT pathway and NF- κ B pathway inhibition.

This is a very well performed study. Overall, this reviewer was impressed with the body of work as presented, and it is certainly of broad significance across neuroscience fields. The experiments as presented are elegantly designed and straightforward in their interpretation towards defining a role for VK on stroke severity. This reviewer has a few minor concerns and clarifications that would benefit this manuscript prior to publication.

Thank you very much for your valuable comments and suggestions from the overall structure of the article and the future application of VK. We take your suggestions.

1. RIGOR guidelines should be followed for all effective translational stroke research. Specifically for this manuscript, female mice and a second stroke model should be used to investigate whether VK is only effective in male mice or MCAO model; longer period time of stroke severity should be demonstrated.

Thank you. We only evaluated the neuroprotective effect of VK (150 and 300 μ g/kg) during the period of acute cerebral ischemia in male mice. However, protective effect of longer period time (such as 14d, 21d, 28d) of VK on stroke severity was not demonstrated in this paper. We are working on the long-term pharmacodynamics of VK on ischemic stroke, such as angiogenesis and synaptic remodeling for effective translational stroke research. We have put your suggestion into the discussion section. The deficiencies of this study. Lines 462-472.

Female mice were not selected because estrogen has a protective effect on cerebral ischemia. In future, We will carry out all these experiments.

2. Systemic toxicity test for VK, such as liver/ kidney, and especial those related to coagulation, platelet function and bleeding should be performed.

Thank you very much. We sought to understand whether VK (0.0091–0.294 μ M) inhibit platelet aggregation platelet aggregation (Supplementary Fig. 2a). Platelet-

activating factor (PAF), arachidonic acid (AA), adenosine diphosphate (ADP), thrombin, and collagen (COL) or saline as inducing agents was employed. ADP-induced platelet aggregation was decreased in VK groups as compared to control group, detailed values are shown in (Supplementary Fig. 2a). Lines 98-103.

Systemic acute toxicity test by peritoneal injection was performed according to the guideline provided by Center for Drug Evaluation and Research in China. Mice were treated intra-peritoneally with single dose of various concentrations (1.5, 6, 24, 96 and 384 mg/kg) of VK and observed for 24 h and no mortality observed. Clinical symptoms (temperature, change in skin, eye color change, general physique, body weight, diarrhea, sedation and organ index) were recorded (Supplementary Table.2). Body weight of mice with treatment VK (24, 96 and 384 mg/kg) continues to decline until day 5 (Supplementary Fig. 2b). The index of brain, liver and kidney had no difference in all treatment dose groups, whereas increased liver index was found (VK, 384 mg/kg) (Supplementary Fig. 2c-f). No significant change was observed for four coagulation during the study (Supplementary Fig. 2g-j). In the histopathological study, it was observed that in all treated groups after 14 days (Supplementary Fig. 3a-d) the organs (brain, liver and kidney) showed no changes at the cellular level in comparison to the control. Karyopyknosis and the cellular swelling of the hepatocyte was moderate in liver of mice treated with VK (96 and 384 mg/kg). Taken together, this finding reveals these dosages of VK are safe to mice.

Lines 104-118.

3. The author does not demonstrate whether VK binds to B1R or B2R, the use of a single antagonist with one in vivo study does not reveal VK-B2R interaction, nor a practical strategy for stroke therapy- such a statement is not accepted.

Thanks. BIAcore experiment were selected to reveal VK-B2R interaction. However, B1R or B2R were seven-span transmembrane protein. We have tried many methods to express and purify it, and very difficult. Finally, to reveal VK-B1R or B2R interaction, the HDOCK server for integrated protein-protein docking was performed as described 47. Known three-dimensional structure of VK and three-dimensional structure of B1R or B2R; we predicted VK-B1R or B2R interaction sites (Fig.8 a-d), and the peptide docking model of VK and B1R with the highest score. The docking summary of the top 10 models were also displayed (Supplementary Fig.7 and Fig.8). Lines 311-316.

4. In Fig. 7c lack of legend. Please check typos throughout the manuscript. For example, line 916 and 972, “1” for “i” , et al.

Thank you. We have carefully checked and revised. For other changes, see highlighted font.

REVIEWERS' COMMENTS:

Reviewer #1 (Remarks to the Author):

The authors addressed the comments and suggestions appropriately and I don't have further concerns.

Reviewer #2 (Remarks to the Author):

This is a revised manuscript by Zhao H et al entitled "Vespa kinin-M, a natural peptide from *Vespa magnifica*, promotes functional recovery in stroke mice". The manuscript is clearly -written, interesting and addresses an important health issue. In generally, the authors have been responsive to the prior reviews, and have modified the manuscript significantly. The manuscript has been improved. In this version, they performed new studies investigating the effects of VK on platelet aggregation function, body weight, coagulation and organ index in mice. There are some new mechanistic analyzing regarding VK-B1R or B2R interaction- The peptide docking model of VK and B1R, VK and B2R. They performed new co-culture and immunofluorescent to show VK 's protective effect on microglia activation.

Sex as a biological variables should be appropriately considered in the experimental design. The authors could do a better job addressing it. Despite this concern, overall this is a good manuscript for publication.

Response to reviewers

Manuscript entitled "Vespakinin-M, a natural peptide from *Vespa magnifica*, promotes functional recovery in stroke mice"

ID: COMMSBIO-21-1583A

Reviewer #1 (Remarks to the Author):

The authors addressed the comments and suggestions appropriately and I don't have further concerns.

Thank you very much. We have benefited a lot from your suggestions and comments.

Reviewer #2 (Remarks to the Author):

This is a revised manuscript by Zhao H et al entitled "Vespakinin-M, a natural peptide from *Vespa magnifica*, promotes functional recovery in stroke mice". The manuscript is clearly -written, interesting and addresses an important health issue. In generally, the authors have been responsive to the prior reviews, and have modified the manuscript significantly. The manuscript has been improved.

In this version, they performed new studies investigating the effects of VK on platelet aggregation function, body weight, coagulation and organ index in mice. There are some new mechanistic analyzing regarding VK-B1R or B2R interaction- The peptide docking model of VK and B1R, VK and B2R. They performed new co-culture and immunofluorescent to show VK 's protective effect on microglia activation.

Sex as a biological variables should be appropriately considered in the experimental design. The authors could do a better job addressing it. Despite this concern, overall this is a good manuscript for publication.

Thanks. Because editors' suggestions and your rigorous audit and broad knowledge improved our manuscript quality. As your consideration, we discussed sex as a

biological variables should be considered.

We only evaluated the neuroprotective effect of VK (150 and 300 $\mu\text{g}/\text{kg}$) during the period of acute cerebral ischemia in male mice. However, efficacy studies were not performed in female mice, and the protective effect of longer period time (such as 14d, 21d, 28 d) of VK on stroke severity was not demonstrated in this study. Human stroke occurs in the context of sex, aging, hypertension, heart disease, diabetes and the use of concomitant medications, sex as a biological variables should be considered in our future research.